# Test-Time Dynamic Image Fusion

**Bing Cao**[1,2]    **Yinan Xia**[1]    **Yi Ding**[1]    **Changqing Zhang**[1,2]    **Qinghua Hu**[1,2*]

[1]College of Intelligence and Computing, Tianjin University, Tianjin, China
[2]Tianjin Key Lab of Machine Learning, Tianjin, China
`{caobing, xyn, ding_yi0731, zhangchangqing, huqinghua}@tju.edu.cn`

## Abstract

The inherent challenge of image fusion lies in capturing the correlation of multi-source images and comprehensively integrating effective information from different sources. Most existing techniques fail to perform dynamic image fusion while notably lacking theoretical guarantees, leading to potential deployment risks in this field. *Is it possible to conduct dynamic image fusion with a clear theoretical justification?* In this paper, we give our solution from a generalization perspective. We proceed to reveal the generalized form of image fusion and derive a new test-time dynamic image fusion paradigm. It provably reduces the upper bound of generalization error. Specifically, we decompose the fused image into multiple components corresponding to its source data. The decomposed components represent the effective information from the source data, thus the gap between them reflects the *Relative Dominability* (RD) of the uni-source data in constructing the fusion image. Theoretically, we prove that the key to reducing generalization error hinges on the negative correlation between the RD-based fusion weight and the uni-source reconstruction loss. Intuitively, RD dynamically highlights the dominant regions of each source and can be naturally converted to the corresponding fusion weight, achieving robust results. Extensive experiments and discussions with in-depth analysis on multiple benchmarks confirm our findings and superiority. Our code is available at `https://github.com/Yinan-Xia/TTD`.

## 1 Introduction

Image fusion jointly integrates complementary information from multiple sources, aiming to generate informative and high-quality fused images. With superior scene representation and enhanced visual perception, image fusion significantly benefits downstream vision tasks [1, 2]. Typically, image fusion can be categorized into multi-modal, multi-exposure, and multi-focus image fusion tasks. Multi-modal image fusion encompasses Visible-Infrared image Fusion (VIF) and Medical Image Fusion (MIF). For VIF [3–5], infrared images effectively highlight thermal targets especially under extreme conditions, while visible images provide texture details and ambient lighting. For MIF [6, 7], different medical imaging modalities emphasize various focal areas, enhancing diagnostic capabilities. Multi-exposure image Fusion (MEF) [8–10] bridges the gap between high dynamic range (HDR) natural scenes and low dynamic range (LDR) pictures, ensuring better detail preservation in varying lighting conditions. Multi-Focus image Fusion (MFF) [11–13] aims to produce all-in-focus images by combining multiple images focused at different depths.

Numerous image fusion methods have been introduced, which can be mainly grouped into traditional techniques and deep learning approaches. Traditional image fusion methods, such as multi-scale decomposition-based models [14, 15] and sparse representation-based methods [16], rely on mathematical transformations to fuse images in the transform domain [17]. In contrast, deep learning-based

---

*Corresponding author.

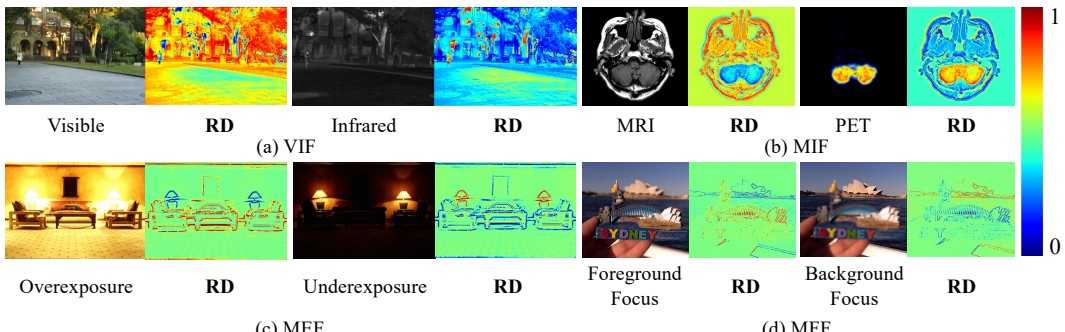

Figure 1: We visualized the Relative Dominablity (RD) of each source on four tasks, which effectively highlights the dominance of uni-source in image fusion.

methods employ data-driven schemes to fuse multi-source images, including convolutional neural network (CNN) based methods [11, 18], generative adversarial network (GAN) based methods [19, 4], and transformer-based methods [20]. The effectiveness of image fusion algorithms hinges on two critical factors: feature extraction [21] and feature fusion [22]. The aforementioned methods strive to achieve high-quality fused images by learning effective uni-source or multi-source feature representations through complex network structures or feature decomposition schemes. However, they often overlook the complexity of the real world, which necessitates dynamic feature fusion.

Recently, some works have highlighted the importance of dynamism in image fusion. For instance, [23] pioneered the combination of image fusion with a Mixture of Experts (MoE), dynamically extracting effective and comprehensive information from the respective modalities. [24] utilized task-specific routing networks to extract task-specific information from different sources with dynamic adapters. Despite their empirically superior fusion performance, these dynamic fusion rules mainly rely on heuristic designs, lacking theoretical guarantees and interpretability. Moreover, they potentially lead to unstable and unreliable fusion results, especially in complex scenarios.

To address these issues, we reveal the generalized form of image fusion and propose a new Test-Time Dynamic(TTD) image fusion paradigm with a theoretical guarantee. Given that the fused image integrates comprehensive information from different sources, it can be obtained by weighting the effective representation of each uni-source. By revisiting the relationship between fusion weights and image fusion losses from the perspective of generalization error [25], we decompose the fused image into multiple uni-source components and formulate the generalization error upper bound of image fusion. Based on generalization theory, we for the first time prove that dynamic image fusion is superior to static image fusion. The key to enhancing generalization lies in the negative correlation between fusion weight and uni-source component reconstruction loss. As fusion models are trained to extract complementary information from each source, the decomposed components represent the effective information from the source data. Thus, the fusion components can be estimated by source data with the fusion model, the losses of which represent the deficiencies of the source in constructing fusion images. Accordingly, we derive a pixel-level Relative Dominablity (RD) as the dynamic fusion weight, which theoretically enhances the generalization of the image fusion model and dynamically highlights the changing dominant regions of different sources as shown in Fig. 1. Extensive experiments on multiple datasets and diverse image fusion tasks demonstrate our superiority. Overall, our contributions can be summarized as follows:

- This paper first theoretically proves the superiority of dynamic image fusion over static image fusion and provides the generalization error upper bound of image fusion by decomposing the fusion image into uni-source components provably. The proposed generalization theory reveals that the key to reducing the upper bound lies in the negative covariance between the fusion weight and uni-source reconstruction loss.

- We proposed a simple but effective test-time dynamic fusion paradigm based on the generalization theory. By taking the uni-source's Relative Dominability as the dynamic fusion weight, we theoretically enhance the generalization of the image fusion model and dynamically emphasize the dominant regions of each source. Notably, our method does not require additional training, fine-tuning, and extra parameters.

- We conduct extensive experiments on multi-modal, multi-exposure, and multi-focus datasets. The superior performance across diverse metrics demonstrates the effectiveness and applicability of our approach. Moreover, an additional exploration of the gradient in constructing fusion weight demonstrates the reasonability of our theory and its expandability.

## 2 Related Works

**Image Fusion** aims to integrate complementary information of diverse source images. For instance, [26] utilize autoencoders to extract multi-source features and fuse them using a designed strategy. GAN-based methods [19] and transformer-based methods [20] also achieved significant progress. [27] introduced the denoising diffusion probabilistic model (DDPM) to image fusion. [28] and [29] achieve considerable fusion performance by decomposing image features into high-frequency and low-frequency components. In addition to these static image fusion methods, [23] used a Mixture of Experts (MoE) to dynamically assign fusion weights, while [24] utilized dynamic adapters to prompt various fusion tasks within a unified model. These approaches mainly focus on obtaining promising feature representations. Although some existing works have explored dynamic image fusion, the lack of theoretical guarantees may result in instability and unreliability in practice.

**Multimodal Dynamic Learning** Although dynamic fusion is not fully studied in existing image fusion works, numerous methods have leveraged multimodal dynamic learning at the decision level [30]. For example, Han et al. [31] assigned dynamic credible fusion weights to each modality at the decision level for robust evidence fusion. Xue et al. [32] employs a Mixture of Experts to integrate the decisions of multiple experts, Zhang et al. [33] combined decision level fusion weight with uncertainty to conduct a credible fusion. Despite the wide exploration of dynamic fusion at the decision level, there is still insufficient research on dynamic fusion at the feature level with theoretical guarantees. In this paper, we focus on the dynamic nature of image fusion, theoretically prove that dynamic fusion is superior to static fusion, and propose a provable feature-level dynamic fusion strategy.

## 3 Method

Given the data from $M$ sources for image fusion, the input samples are denoted as $\{x = x^{(m)} \mid m = 1, 2, \ldots, M\}$, where $x^{(m)}$ represents the input from the $m$-th source. Let $f$ be the image fusion model, comprising both encoders and decoders. Define $E = \{E^{(m)}(\cdot) \mid m = 1, 2, \ldots, M\}$ as the set of encoders within the image fusion network, where $E^{(m)}(\cdot)$ is the encoder for the $m$-th source. In early fusion, the encoders in $E$ are constant mapping functions, meaning that multi-source images are combined at the image level. Let $D(\cdot)$ denote the decoder in the image fusion network, and let $\omega = \{\omega^{(m)} \in \mathbb{R}^{H \times W} \mid m = 1, 2, \ldots, M\}$ represent the set of image fusion weights. Consequently, the fused image $I_F$ can be expressed as:

$$I_F = D\Big( \sum_{m=1}^{M} \omega^{(m)} E^{(m)}(x^{(m)}) \Big). \tag{1}$$

Additionally, we define the loss function for image fusion tasks, where $\| \cdot \|$ represents any distance norm. The loss function is given by:

$$\ell(I_F, x) = \sum_{m=1}^{M} \|I_F - x^{(m)}\|. \tag{2}$$

**Generalization Error Upper Bound.** In machine learning, the concept of Generalization Error Upper Bound refers to the theoretical limit on a model's performance when applied to unseen data ($\mathcal{D}$) [34]. A smaller upper bound indicates better-expected performance on data from an unknown distribution. For image fusion tasks, the Generalization Error (GError) of a fusion model $f$ can be defined as:

$$\text{GError}(f) = \mathbb{E}_{x \sim \mathcal{D}}[\ell(f(x), x)]. \tag{3}$$

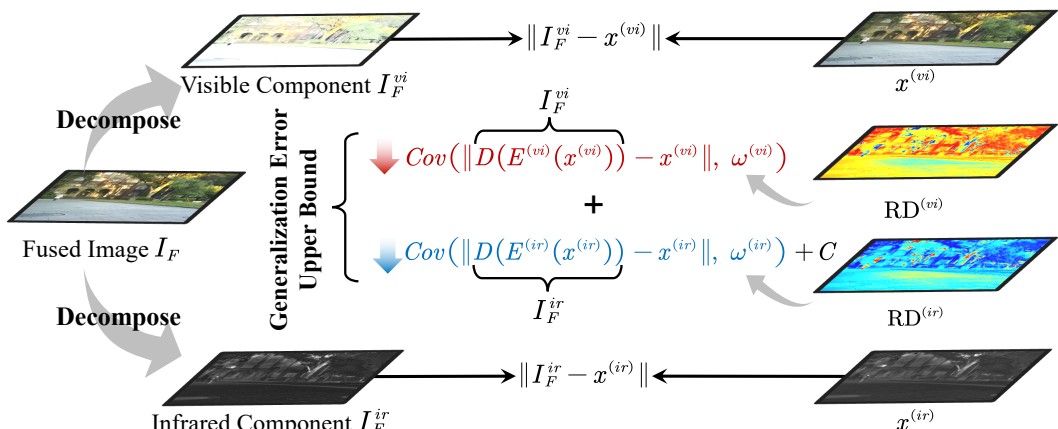

Figure 2: The framework of our TTD. Deriving from the generalization theory, we decompose fused images into uni-source components and find the key to reducing generalization error upper bound is the negative correlation between the fusion weight and reconstruction loss. Accordingly, we propose pixel-wise Relative Dominablity (RD) for each source, which is negatively correlation with the reconstruction loss and highlights the dominant regions of uni-source in constructing fusion images.

Considering the GError of image fusion model [35–37], $\ell(I_F, x)$ can be further deduced as $\ell(I_F, x) \leq \sum_{m=1}^{M} \| \sum_{i=1}^{M} \omega^{(i)} \cdot D(E^{(i)}(x^{(i)})) - x^{(m)}\|$. Therefore, the fused image $I_F$ is decomposed into $M$ uni-source components $\{D\left(E^{(i)}\left(x^{(i)}\right)\right) | i = 1..M\}$. Based on Eq. (3) and (8), we have:

**Theorem 3.1** *(Decomposition of Generalization Error). The GError for multi-source image fusion model f can be decomposed into a linear combination of each uni-source component reconstruction loss under the condition that $\sum_{m=1}^{M} \omega^{(m)} = 1$, the detailed proof is given in Appendix A.1:*

$$\text{GError}(f) = \mathbb{E}_{x \sim \mathcal{D}}[\sum_{m=1}^{M} \| \sum_{i=1}^{M} D(\omega^{(i)} \cdot E^{(i)}(x^{(i)})) - x^{(m)}\|]$$

$$\leq \frac{1}{M} \sum_{m=1}^{M} \mathbb{E}\big[(2M-1)\|D(E^{(m)}(x^{(m)})) - x^{(m)}\| + (M-1)\sum_{i \neq m}^{M} \|D(E^{(i)}(x^{(i)})) - x^{(m)}\|\big]$$

$$+ \sum_{m=1}^{M} Cov\big(\omega^{(m)}, \| \underbrace{D\left(E^{(m)}\left(x^{(m)}\right)\right)}_{\text{uni-source component}} - x^{(m)}\|\big). \tag{4}$$

Let $\ell^{(m)} = \|D\left(E^{(m)}\left(x^{(m)}\right)\right) - x^{(m)}\|$, which represents the reconstruction loss between a uni-source component and its corresponding uni-source image. The term $Cov(\omega^{(m)}, \ell^{(m)})$ denotes the covariance between $\omega^{(m)}$ and $\ell^{(m)}$. The essence of reducing generalization error lies in achieving the lowest possible fusion loss. By leveraging the triangular inequality properties of distance norms within the fusion loss, we can deduce that the GError is bounded by the covariance term and the distance between each uni-source component and its source image. It is noteworthy that $f(x^{(m)})$ remains constant during the test phase, emphasizing that the pivotal factor in reducing Generalization Error Upper Bound (GEB) lies in the covariance between $\omega^{(m)}$ and $\ell^{(m)}$.

**Superiority of Dynamic Image Fusion over Static Image Fusion.** Most existing image fusion approaches reduce GEB by minimizing $\ell^{(m)}$, indicating an effort to enhance the quality of uni-source feature representations. However, they often overlook the intrinsic significance of fusion weight $\omega^{(m)}$. Fusion strategies employed in static image fusion encompass methods such as maximum, minimum, addition, $\ell_1$-norm, etc. Nevertheless, none of these fusion weights exhibit a correlation with uni-source reconstruction loss, i.e. $Cov(\omega_{static}^{(m)}, \ell^{(m)}) = 0$. During the test phase, $\ell^{(m)}$ remains constant. If we have: $Cov(\omega_{dynamic}^{(m)}, \ell^{(m)}) < 0$ for all source images, we can derive the conclusion:

$$\text{GEB}_{dynamic} < \text{GEB}_{static}. \tag{5}$$

This indicates that for a well-trained image fusion model, a dynamic fusion strategy can bring better generalization than a static fusion strategy.

**Relative Dominablity.** Recalling the Eq. (4), the negative correlation between fusion weight and the reconstruction loss $\ell^{(m)}$ provably reduces the generalization error upper bound. Therefore, we introduce a pixel-level Relative Dominablity (RD) as the fusion weight for each source, which exhibits a negative correlation with the reconstruction loss of the corresponding fusion component. Since fusion models are trained to extract complementary information from each source, the decomposed components of fusion images represent the effective information from the source data. Thus, the uni-source components can be estimated from source data using the fusion model, with the losses representing the deficiencies of the source in constructing fusion images. For instance, in a given region, the larger the pixel-wise fusion loss between the reconstructed component and its corresponding uni-source image, the smaller its contribution to image fusion. Intuitively, using RD as the dynamic fusion weight can capture the dominance of each source in image fusion and enhance its advantages in constructing fusion images. Theoretically, according to Thm. 3.1, negatively correlated with the pixel-wise fusion loss, RD effectively demonstrates the dominance of each source. Notably, considering the relative nature of multi-source image fusion, the sum of the RDs of different sources for the same pixel should be one. Consequently, by establishing a negative correlation with the loss and implementing normalization, we can obtain the Relative Dominablity of each source for a certain sample as follows:

$$\omega^{(m)} = \text{RD}^{(m)} = Softmax(e^{-\ell^{(m)}}). \tag{6}$$

In addition, we present the algorithm and test pipeline of our dynamic fusion strategy in Appendix B.1.

## 4  Experiments

### 4.1  Experimental Setup

**Datasets.** We evaluate our proposed method on four image fusion tasks: Visible-Infrared Fusion (VIF), Medical Image Fusion (MIF), Multi-Exposure Fusion (MEF), and Multi-Focus Fusion (MFF). ∘ VIF: For VIF tasks, we conduct experiments on two datasets: LLVIP [38] and MSRS [17]. For LLVIP datasets, we randomly select 70 samples from the test set for evaluation. ∘ MIF: We conduct experiments on the Harvard Medical Image Dataset, following the test setting in [29]. ∘ MEF: Following the setting in [24], we verified the performance of our method on MEFB [39] dataset. ∘ MFF: For the MFF task, we evaluate our method on MFI-WHU datasets [40], following the test protocol in [24]. As a test-time adaption approach, TTD performs adaptive fusion solely during testing, without additional training and training data.

**Competing Methods.** For VIF and MIF tasks, we evaluated 12 state-of-the-art methods, encompassing both DenseFuse [26], CDDFuse [29], U2Fusion [22], DDFM [27], DeFusion [41], PIAFusion [17], DIVFusion [42], MUFusion [43], IFCNN [18], and SwinFuse [20], and TC-MoA [24]. For MEF and MFF tasks, we compared our methods with general image fusion methods and task-specific image fusion methods. Notably, among these methods, only DDFM is training-free, and other methods are all pre-trained on VIF datasets. In experiments, we apply our TTD to CDDFuse (CDDFusion+TTD), PIAFusion (PIAFusion+TTD), and IFCNN (IFCNN+TTD), separately. Our experiments are conducted on Huawei Atlas 800 Training Server with CANN and NVIDIA RTX A6000 GPU.

**Metrics.** We selected several evaluation metrics from three aspects [44], including ∘ *information theory*: entropy (EN) [45], cross entropy (CE), the sum of the correlations of differences (SCD) [46], ∘ *image feature*: standard deviation (SD), average gradient (AG) [47], edge intensity (EI) and spatial frequency (SF) [48], and ∘ *structural similarity*: structural similarity (SSIM) [49].

### 4.2  Quantitative Comparisons

**Visible-Infrared Fusion.** Tab. 1 reports the performance of competing approaches and TTD-applied methods on LLVIP and MSRS datasets for 7 metrics. Notably, by applying our TTD, the previous methods have improved on most of the indicators. Also, our TTD strategy outperforms other traditional static methods, training-free method DDFM, and data-driven dynamic strategy TC-MoA, achieving the SoTA performance on most metrics. Moreover, with particularly high values in SD, AG, EI, and SF, our TTD ensures that fusion images maintain exceptional contrast and detailed texture, highlighting its efficacy in preserving quality. The outstanding performance on EN and SCD indicates

Table 1: Quantitative performance comparison of different fusion strategies on visible-infrared datasets. The 'TTD' suffix and gray background indicates our method is applied to this baseline. The **red** and **blue** represent the best and second-best result respectively. The **bold** indicates the baseline w/ TTD performance better than that w/o TTD. We used △ to illustrate the amount of improvement our TTD method achieved compared to the baseline.

| Method | LLVIP Dataset | | | | | | | MSRS Dataset | | | | | | |
|---|---|---|---|---|---|---|---|---|---|---|---|---|---|---|
| | EN↑ | SD↑ | AG↑ | EI↑ | SF↑ | SCD↑ | CE↓ | EN↑ | SD↑ | AG↑ | EI↑ | SF↑ | SCD↑ | CE↓ |
| Densefuse [26] | 6.83 | 33.98 | 3.62 | 8.80 | 12.27 | 1.24 | 8.13 | 5.93 | 23.55 | 2.05 | 5.42 | 6.02 | 1.25 | 7.75 |
| U2Fusion [22] | 6.64 | 35.83 | 4.13 | 10.56 | 13.70 | 1.27 | 9.14 | 5.21 | 22.67 | 2.51 | 6.70 | 8.06 | 1.15 | 12.54 |
| DeFusion [41] | 7.21 | 42.91 | 3.80 | 9.76 | 11.99 | 1.21 | 7.83 | 6.38 | 35.43 | 2.64 | 7.12 | 8.15 | 1.27 | 7.55 |
| SwinFuse [20] | 5.84 | 40.95 | 3.58 | 9.02 | 15.38 | 1.27 | 8.51 | 4.24 | 29.72 | 1.93 | 5.12 | 9.47 | 1.03 | 8.93 |
| MUFusion [43] | 7.29 | 50.09 | 4.96 | 13.38 | 13.29 | 1.38 | 7.57 | 6.09 | 31.81 | 3.46 | 9.54 | 9.77 | 1.33 | 6.87 |
| DDFM [27] | 6.34 | 32.31 | 3.25 | 7.93 | 11.71 | 1.07 | 8.66 | 5.76 | 22.94 | 2.01 | 5.28 | 6.44 | 1.22 | 7.56 |
| TC-MoA [24] | 7.40 | 48.92 | 2.76 | 7.47 | 9.78 | 1.40 | 7.83 | 6.49 | 35.60 | 3.12 | 8.99 | 10.77 | 1.33 | 7.12 |
| IFCNN [18] | 6.95 | 37.75 | 5.18 | 13.13 | 18.18 | 1.32 | 7.82 | 6.07 | 26.99 | 3.44 | 8.99 | 10.77 | 1.33 | 7.12 |
| IFCNN+TTD | 6.98 | 38.99 | 5.48 | 13.92 | 19.40 | 1.34 | 7.79 | 6.09 | 28.09 | 3.58 | 9.39 | 11.46 | 1.35 | 7.10 |
| Improve | △0.03 | △1.24 | △0.30 | △0.79 | △1.22 | △0.02 | △0.03 | △0.02 | △1.10 | △0.14 | △0.40 | △0.69 | △0.02 | △0.02 |
| CDDFuse [29] | 7.36 | 50.90 | 4.99 | 12.68 | 18.26 | 1.62 | 7.79 | 6.71 | 43.38 | 3.78 | 10.08 | 11.57 | 1.60 | 6.92 |
| CDDFuse+TTD | 7.34 | 53.88 | 5.54 | 14.07 | 20.17 | 1.58 | 7.83 | 6.64 | 43.78 | 3.97 | 10.54 | 12.67 | 1.58 | 6.95 |
| Improve | ▽0.02 | △2.98 | △0.55 | △1.39 | △1.91 | ▽0.04 | △0.16 | ▽0.07 | △0.40 | △0.19 | △0.46 | △1.10 | ▽0.02 | ▽0.03 |
| PIAFusion [17] | 7.39 | 52.12 | 5.77 | 14.81 | 19.59 | 1.59 | 7.72 | 6.64 | 45.34 | 3.95 | 10.57 | 12.12 | 1.70 | 6.93 |
| PIAFusion+TTD | 7.42 | 56.14 | 5.90 | 15.15 | 20.29 | 1.64 | 7.65 | 6.62 | 48.26 | 4.18 | 11.18 | 12.99 | 1.52 | 6.92 |
| Improve | △0.04 | △4.02 | △0.13 | △0.34 | △0.70 | △0.05 | △0.07 | ▽0.02 | △2.92 | △0.23 | △0.61 | △0.87 | ▽0.18 | △0.01 |

Table 2: Quantitative comparison on MFI-WHU dataset in MFF task and MEFB dataset in MEF task.

| Method | MEFB Dataset | | | | | | Method | MFI-WHU Dataset | | | | | |
|---|---|---|---|---|---|---|---|---|---|---|---|---|---|
| | SD↑ | EI↑ | EN↑ | AG↑ | SF↑ | CE↓ | | SD↑ | EI↑ | EN↑ | AG↑ | SF↑ | CE↓ |
| PMGI [50] | 62.36 | 13.33 | 7.25 | 5.35 | 18.60 | 9.57 | PMGI [50] | 44.64 | 11.03 | 7.10 | 4.20 | 11.36 | 10.00 |
| U2Fusion [22] | 52.27 | 12.06 | 6.93 | 4.65 | 15.37 | 13.66 | U2Fusion [22] | 54.38 | 18.24 | 7.32 | 7.03 | 19.10 | 7.95 |
| DeFusion [41] | 46.85 | 10.51 | 6.78 | 4.07 | 13.48 | 13.55 | DeFusion [41] | 50.78 | 12.23 | 7.29 | 4.65 | 12.56 | 7.42 |
| TC-MoA [24] | 48.91 | 12.13 | 7.06 | 4.77 | 15.56 | 11.91 | TC-MoA [24] | 53.35 | 17.31 | 7.36 | 6.83 | 20.56 | 7.40 |
| Deepfuse [9] | 48.29 | 8.74 | 6.97 | 3.34 | 9.90 | 12.33 | DRPL [12] | 53.88 | 18.93 | 7.38 | 7.66 | 23.66 | 7.38 |
| DEM [51] | 52.35 | 13.60 | 7.32 | 5.46 | 18.85 | 11.68 | ECNN [13] | 53.79 | 18.77 | 7.38 | 7.59 | 23.51 | 7.38 |
| DSIFT_EF [52] | 50.65 | 12.54 | 7.36 | 5.00 | 17.18 | 12.47 | GCF [53] | 53.78 | 18.81 | 7.38 | 7.60 | 23.56 | 7.38 |
| MEFAW [54] | 48.31 | 12.41 | 7.22 | 4.95 | 16.86 | 11.74 | GFDF [55] | 53.72 | 18.70 | 7.38 | 7.55 | 23.38 | 7.39 |
| MEFCNN [56] | 51.29 | 12.23 | 7.24 | 4.88 | 16.89 | 13.15 | MADCNN [57] | 53.85 | 18.82 | 7.38 | 7.59 | 23.38 | 7.38 |
| MEFOpt [58] | 49.21 | 14.06 | 7.18 | 5.63 | 19.27 | 12.09 | PCANet [59] | 53.73 | 18.61 | 7.37 | 7.51 | 23.33 | 7.38 |
| GALFusion [60] | 50.39 | 9.53 | 6.95 | 3.78 | 13.04 | 13.12 | SESF [61] | 53.85 | 18.75 | 7.38 | 7.57 | 23.57 | 7.38 |
| HoLoCo [62] | 52.83 | 12.27 | 7.19 | 4.65 | 13.56 | 13.63 | TF [63] | 53.71 | 18.72 | 7.38 | 7.57 | 23.41 | 7.38 |
| IFCNN [18] | 51.78 | 14.48 | 7.06 | 5.85 | 20.41 | 11.60 | IFCNN | 54.02 | 18.71 | 7.38 | 7.55 | 22.85 | 7.40 |
| IFCNN+TTD | 52.86 | 15.94 | 7.10 | 6.38 | 21.99 | 11.41 | IFCNN+TTD | 54.22 | 19.10 | 7.39 | 7.70 | 23.51 | 7.40 |
| Improve | △1.08 | △1.46 | △0.04 | △0.53 | △1.58 | △0.19 | Improved | △0.20 | △0.39 | △0.01 | △0.15 | △0.66 | △0.0 |

our fusion results embed more information and contain abundant edge information from the source images. Although our approach is a test time adaptation strategy that does not require training, the designed dynamic weights based on theoretical principles have led to outstanding fusion performance, achieving SoTA results on VIF tasks.

**Medical Image Fusion.** We report the comparison results on three MIF scenarios: MRI-CT, MRI-PET, and MRI-SPECT. As depicted in Tab. 3, 6 and Tab. 7 in Appendix C.3, our method yields competitive performance on seven evaluation metrics. Specifically, our TTD enhances EN, AG, and SSIM, indicating ample gradient information and structural details in the fusion results. The significant improvements in SD, EI, and SF highlight our high definition and texture quality compared with the competing methods, making it exceptionally competitive.

**Multi-Exposure and Multi-Focus Image Fusion.** In the comparisons on MEF and MFF tasks, we applied our TTD to the general fusion method IFCNN. We compared it with other general fusion methods and task-specific fusion methods. As depicted in Tab. 2, TTD outperforms existing general fusion methods and task-specific methods in terms of SD, EI, AG, and SF. This indicates that TTD produces fused images with clear, abundant edges and exceptional sharpness. Furthermore, the superiority in EN and CE suggests that our TTD enables the baseline to preserve more advantages from different sources.

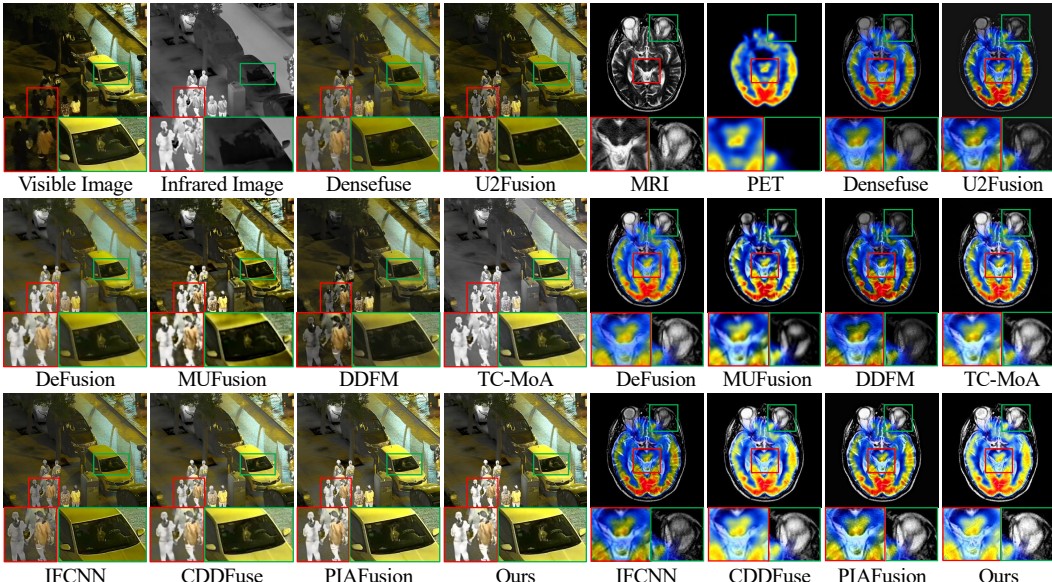

| Visible Image | Infrared Image | Densefuse | U2Fusion | MRI | PET | Densefuse | U2Fusion |

| DeFusion | MUFusion | DDFM | TC-MoA | DeFusion | MUFusion | DDFM | TC-MoA |

| IFCNN | CDDFuse | PIAFusion | Ours | IFCNN | CDDFuse | PIAFusion | Ours |

Figure 3: (a) On the VIF task, our TTD produces fused images that retain more multi-source information compared with existing approaches. (b) On the MIF task, our method improves the contrast of the fused image and preserves more details from the source image.

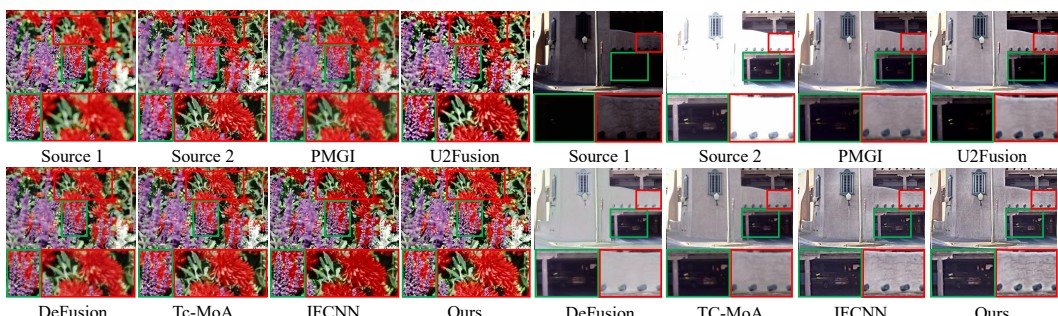

| Source 1 | Source 2 | PMGI | U2Fusion | Source 1 | Source 2 | PMGI | U2Fusion |

| DeFusion | Tc-MoA | IFCNN | Ours | DeFusion | TC-MoA | IFCNN | Ours |

Figure 4: The comparison of fusion results on MEF and MFF tasks. (a) On the MFF task, our method retains the color and clarity of the original image better. (b) On the MEF task, our TTD ensures better detail preservation in varying lighting conditions.

### 4.3 Qualitative Comparisons

**Visible-Infrared Fusion.** As shown in Fig. 3 and Fig. 8 in Appendix C.1, compared with existing methods on the LLVIP dataset, our TTD effectively combines comprehensive information from different sources, leading to a significant visual performance. Specifically, the fusion result not only preserves the texture details and edge information of visible images but also incorporates high-quality thermal imaging contrast of infrared images. Additionally, as mentioned in the qualitative analysis, our fusion images exhibit high fidelity and clear contrasts, showing consistent superiority in terms of image quality. These experimental results demonstrate the effectiveness of our TTD.

**Medical Image Fusion.** For the MIF task, we present qualitative comparisons of the MRI-PET fusion. As shown in Fig. 3 and Fig. 8 in Appendix C.1, it is clear that fusion images generated by our method preserve a substantial amount of structural information. Notably, our method maintains a significant portion of excellent soft tissue contrast details from MRI images and combines the quantitative physiologic information of PET images. With our TTD, the overall structural details and sharpness of the fused image are significantly enhanced. Moreover, in the regions where the high-intensity color areas of the PET image overlap with the structural information of the MRI, the detailed information from the original images is well preserved and highlighted in the fused image.

**Multi-Exposure and Multi-Focus Image Fusion.** We also provide a comparison of fusion results for the MEF and MFF tasks in Fig. 4 and Fig. 8 in Appendix C.1. Notably, our TTD significantly enhances the clarity and sharpness of texture details. Obviously, after applying our TTD, the fused images on the MEF task exhibit higher clarity in both the foreground and background. For the MFF task, our method accurately utilizes the effective regions from both underexposed and overexposed images. Compared to other methods, our fusion results achieve more precise exposure and rich texture details, such as the cars in the garage and the textures on the walls. Additionally, our fusion method retains high-fidelity colors that are closer to the original images.

Apart from the comparisons with the existing methods, we provided more ablated comparisons with baselines in Fig. 9 of Appendix C.2.

## 5 Discussion

### 5.1 Is Negative Correlation Help?

The negative correlation between RD and $\ell^{(m)}$ is derived from Eq. (4) to reduce the generalization error of the image fusion model. To further validate the effectiveness of the theoretical guarantee, we compare it with a contrast setting: using a new fusion weight, which is positively correlated (PC) to $\ell^{(m)}$, to perform image fusion.

As shown in the correlation comparison in Fig. 5 (b), loss-positive-correlated weights (yellow line), that conflict with our theory, lead to a decreased performance compared with static fusion (green line). As a comparison, the results of the loss-negative-correlated fusion strategy (red line), exhibit superior performance compared with both static image fusion and positively correlated fusion strategy. These experiments verify that the proposed negative correlation setting can explicitly reduce the generalization error, demonstrating the reasonability of the proposed TTD image fusion model.

### 5.2 Relative Dominability

In this paper, we introduce the pixel-level Relative Dominablity, which indicates the advantages of each source. Treating the Relative Dominablity as dynamic fusion weight, TTD achieves an adaptive and interpretable image fusion. We provide visualizations of each source's pixel-level Relative Dominability obtained using CDD+TTD for VIF and MIF tasks, and IFCNN+TTD for MEF and MFF tasks.

**Visible-Infrared Fusion.** As shown in Fig. 1, it can be observed that Relative Dominablity (RD) accurately reflects the dominance of each source: in visible images, well-lit and properly exposed bright areas contain abundant brightness information, and areas like digits and characters exhibit rich texture details. In contrast, infrared images provide thermal imaging information for areas and objects in shadow that visible images cannot capture due to visual obstacles. The proposed RD effectively captures the advantageous regions of different source images and assigns larger weights to these regions, thereby achieving more reliable fusion results.

**Medical Image Fusion.** We visualize RDs in the MRI-PET dataset. Similar findings are also apparent in the MIF task. In PET images, bright regions indicate areas of malignant cell proliferation, while MRI contains more structural information. As shown in Fig. 1, the RDs of PET stand out in the bright areas while MRI's highlights the structural information. Guided by RD, TTD emphasizes potential lesion areas while preserving the structural information of these areas effectively.

**Multi-Exposure and Multi-Focus Image Fusion.** For MEF and MFF tasks, the ideal outcome is that the fused images contain properly exposed or precisely focused regions from each uni-source. As shown in the visualized RD map in Fig. 1, our TTD can effectively capture the dominant areas in different sources and assign higher RD values, i.e. dynamic fusion weight, to these regions.

**Downstream Tasks.** To validate the effectiveness of our RD on downstream tasks, we compared our TTD with the baseline on an object detection task. Detailed results are given in Appendix C.5.

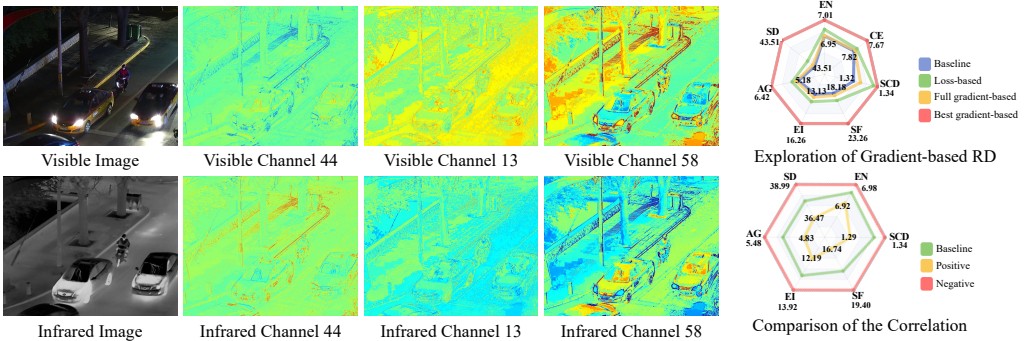

| Visible Image | Visible Channel 44 | Visible Channel 13 | Visible Channel 58 | Exploration of Gradient-based RD |
| Infrared Image | Infrared Channel 44 | Infrared Channel 13 | Infrared Channel 58 | Comparison of the Correlation |

(a) The visualization of RDs obtained by gradient maps of different channels     (b) The radar charts of experiments in discussion

Figure 5: (a) The visualization of RDs obtained by gradient maps of different channels. The 44th gradient map provides wrong dominance information, and the 13th gradient map offers insignificant information, while the 58th gradient map performs the proper advantages of the two source images. (b) The radar chart of the gradient-based RD experiment (upper) and the validation of the negative correlation (below).

Overall, with the integration of our TTD, the baseline model gains the ability to perceive dominant information dynamically. Therefore, this interpretable plug-and-play test-time dynamic adaptation fusion strategy can further improve the performance of the existing state-of-the-art methods. This further validates the effectiveness of RD in our TTD.

## 5.3 Gradient-based Relative Dominability

In our TTD, the proposed pixel-level fusion weight is computed by pixel-level loss. However, some numeric losses are limited in directly obtaining the pixel-level weights, making it hard to integrate with TTD flexibly. To overcome this dilemma, we extend our TTD to a more general form and construct gradient-based Relative Dominability through any fusion losses for a more fine-grained and robust image fusion.

Recalling our optimization objective of the generalization error bound, we aim for a negative correlation between the weights and losses of the same modality, i.e., establishing a correlation between losses and weights. Inspired by this positive correlation between loss value and the absolute value of its gradient for features with any convex loss function, our TTD can be further extended to a gradient-based dynamic fusion weight.

Specifically, we first calculate the absolute value of gradients $|G^{(m)}| \in \mathbb{R}^{H \times W \times C}$ of each uni-source feature. As a test-time adaption approach, TTD does not update the network parameters, meaning that the unimodal feature space remains fixed for the same baseline. For the same task scenarios, the feature patterns tend to be similar. Therefore, we can empirically select the gradient channels that well represent the advantage areas to compute RDs. Also, as illustrated in Fig. 5 (a), gradient maps of some channels (such as the 44th and the 13th channels) lack significant useful information and fail to capture advantageous regions in the original images. Therefore, we select the gradient map $|g^{(m)}| \in \mathbb{R}^{H \times W}$ among $C$ channels that best represent the dominance of the uni-source empirically. By replacing $\ell^{(m)}$, we can obtain the RD and the dynamic fusion weight as follows:

$$\omega^{(m)} = \text{RD}^{(m)} = Softmax(e^{-|g|^{(m)}}). \tag{7}$$

We have also conducted comparisons of loss-based TTD, full gradient-based TTD, and best gradient-based TTD on IFCNN. For the best gradient-based TTD, we choose the 58-th gradient map to obtain the fusion weight. The results of gradient-based RD in Fig. 5 (b) demonstrate that full gradient (yellow line) may bring wrong or useless dominance information to fusion weights, leading to worse performance compared with loss-based TTD (green line). However, by selecting the empirically best gradient map (red line), our TTD provides more fine-grained dominance information compared to the global loss maps, achieving more detailed dynamic fusion with better performance.

# 6  Conclusion

Image fusion aims to integrate effective information from multiple sources. Despite numerous methods being proposed, research on dynamic fusion and its theoretical guarantees remains significantly lacking. To address these issues, we derive from a generalized form of image fusion and introduce a new Test-Time Dynamic (TTD) image fusion paradigm with a theoretical guarantee. From the perspective of generalization error, we reveal that reducing generalization error hinges on the negative correlation between the fusion weight and the uni-source component reconstruction loss. Here the uni-source components are decomposed from fusion images, reflecting the effective information of the corresponding source image in constructing fusion images. Accordingly, we propose a pixel-level Relative Dominablity (RD) as the dynamic fusion weight, which theoretically enhances the generalization of the image fusion model and dynamically highlights the changing dominant regions of different sources. Comprehensive experiments with in-depth analysis validate our superiority. We believe the proposed TTD paradigm is an inspirational development that can benefit the community and address the theoretical gap in image fusion research.

## Acknowledgements

This work was sponsored by the National Natural Science Foundation of China (No.s U23B2049, 62476198, 62376193, 62106171, 61925602), and CCF-Baidu Open Fund. This work was also sponsored by CAAI-CANN Open Fund, developed on OpenI Community. *Yinan Xia* and *Yi Ding* contributed equally to this work. The authors thank anonymous peer reviewers for their helpful suggestions.

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

# Appendix

## A    Proof

### A.1    Proof of Theorem 3.1

*Proof.* By leveraging the properties of convex functions, the loss function can be derived to the following inequality when $\sum_{i=1}^{M} \omega^{(i)} = 1$:

$$\ell(I_F, x) = \sum_{m=1}^{M} \|I_F - x^{(m)}\| = \sum_{m=1}^{M} \left( \| \sum_{i=1}^{M} D(\omega^{(i)} \cdot E^{(i)}(x^{(i)})) - x^{(m)} \| \right)$$

$$\leq \sum_{m=1}^{M} \left( \| \sum_{i=1}^{M} \omega^{(i)} \cdot D(E^{(i)}(x^{(i)})) - x^{(m)} \| \right)$$

$$= \sum_{m=1}^{M} \left( \sum_{i=1}^{M} \omega^{(i)} \cdot \| D(E^{(i)}(x^{(i)})) - x^{(m)} \| \right). \tag{8}$$

Therefore, the fused image $I_F$ can be decomposed to M uni-source components. By taking the expectation on both sides of the inequality, we can derive the generalization error GError($f$) of the model on the unknown distribution $\mathcal{D}$.

$$\text{GError}(f) \leq \mathbb{E}_{x \sim \mathcal{D}} \left[ \sum_{m=1}^{M} \left( \sum_{i=1}^{M} \omega^{(i)} \cdot \| D(E^{(i)}(x^{(i)})) - x^{(m)} \| \right) \right]. \tag{9}$$

For simplicity, we use $\ell \left( x_{com}^{(i)}, x^{(j)} \right)$ to denote $\| D(E^{(i)}(x^{(i)})) - x^{(j)} \|$. Consequently, leveraging the triangle inequality for norms and the fact that $\sum_{m=1}^{M} \omega^{(m)} = 1$, we have:

$$\text{GError}(f) \leq \sum_{m=1}^{M} \mathbb{E}_{x \sim \mathcal{D}} \left[ \sum_{i=1}^{M} \omega^{(i)} \cdot \ell \left( x_{com}^{(i)}, x^{(m)} \right) \right]$$

$$= \sum_{m=1}^{M} \mathbb{E}_{x \sim \mathcal{D}} \left[ \omega^{(m)} \cdot \ell \left( x_{com}^{(m)}, x^{(m)} \right) + \sum_{i=1, i \neq m}^{M} \omega^{(i)} \cdot \ell \left( x_{com}^{(i)}, x^{(m)} \right) \right]$$

$$= \frac{1}{M} \cdot \sum_{m=1}^{M} \mathbb{E}_{x \sim \mathcal{D}} \left[ \sum_{i=1, i \neq m}^{M} (1 - \sum_{j=1, j \neq i}^{M} \omega^{(j)}) \ell \left( x_{com}^{(i)}, x^{(m)} \right) + (M-1)\omega^{(i)} \cdot \ell \left( x_{com}^{(i)}, x^{(m)} \right) \right]$$

$$+ \sum_{m=1}^{M} \mathbb{E}_{x \sim \mathcal{D}} \left[ \omega^{(m)} \right] \mathbb{E}_{x \sim \mathcal{D}} \left[ \ell \left( x_{com}^{(m)}, x^{(m)} \right) \right] + Cov \left( \omega^{(m)}, \ell \left( x_{com}^{(m)}, x^{(m)} \right) \right)$$

$$\leq \frac{1}{M} \cdot \sum_{m=1}^{M} \mathbb{E}_{x \sim \mathcal{D}} \left[ \sum_{i=1, i \neq m}^{M} \ell \left( x_{com}^{(i)}, x^{(m)} \right) + \ell \left( x_{com}^{(i)}, x^{(m)} \right) \left( (M-1)\omega^{(i)} - \sum_{j=1, j \neq i}^{M} \omega^{(j)} \right) \right]$$

$$+ \sum_{m=1}^{M} \mathbb{E}_{x \sim \mathcal{D}} \left[ \ell \left( x_{com}^{(m)}, x^{(m)} \right) \right] + Cov \left( \omega^{(m)}, \ell \left( x_{com}^{(m)}, x^{(m)} \right) \right)$$

$$\leq \frac{M-1}{M} \cdot \sum_{m=1}^{M} \mathbb{E}_{x \sim \mathcal{D}} \left[ \ell \left( x_{com}^{(m)}, x^{(m)} \right) + \sum_{i=1, i \neq m}^{M} \ell \left( x_{com}^{(i)}, x^{(m)} \right) \right] + \sum_{m=1}^{M} \mathbb{E}_{x \sim \mathcal{D}} \left[ \ell \left( x_{com}^{(m)}, x^{(m)} \right) \right]$$

$$+ \sum_{m=1}^{M} \left[ Cov \left( \omega^{(m)}, \ell \left( x_{com}^{(m)}, x^{(m)} \right) \right) \right]$$

$$= \sum_{m=1}^{M} \left( Cov \left( \omega^{(m)}, \ell \left( x_{com}^{(m)}, x^{(m)} \right) \right) + \mathbb{E}_{x \sim \mathcal{D}} \left[ \underbrace{\frac{2M-1}{M} \ell \left( x_{com}^{(m)}, x^{(m)} \right) + \frac{M-1}{M} \sum_{i=1, i \neq m}^{M} \ell \left( x_{com}^{(i)}, x^{(m)} \right)}_{\text{constant}} \right] \right)$$

$$= \sum_{m=1}^{M} Cov \left( \omega^{(m)}, \| \underbrace{D \left( E^{(m)} \left( x^{(m)} \right) \right)}_{\text{uni-source component}} - x^{(m)} \| \right) + C. \tag{10}$$

Table 3: Quantitative comparison on MRI-CT dataset in medical image fusion task.

| Method | MRI-CT Dataset | | | | | | |
|---|---|---|---|---|---|---|---|
| | EN↑ | SD↑ | AG↑ | EI↑ | SF↑ | SSIM↑ | CE↓ |
| Densefuse [26] | 4.51 | 57.06 | 4.73 | 12.19 | 19.37 | 1.49 | 4.95 |
| U2Fusion [22] | 4.87 | 53.80 | 6.23 | 16.20 | 22.48 | 0.57 | 17.45 |
| DeFusion [41] | 4.60 | 66.17 | 5.38 | 14.22 | 21.55 | 1.49 | 4.63 |
| SwinFuse [20] | 3.94 | 72.61 | 7.04 | 17.66 | 35.96 | 1.45 | 4.71 |
| MUFusion [43] | 4.74 | 79.41 | 6.44 | 17.62 | 21.04 | 1.38 | 4.91 |
| DDFM [27] | 4.59 | 62.55 | 5.48 | 14.03 | 23.77 | 1.47 | 4.75 |
| TC-MoA [24] | 5.37 | 78.62 | 7.01 | 18.62 | 26.18 | 1.42 | 5.20 |
| IFCNN [18] | 4.62 | 61.98 | 7.86 | 19.92 | 31.06 | 1.49 | 4.72 |
| IFCNN+TTD | 4.66 | 66.14 | 8.27 | 21.11 | 32.63 | 1.50 | 4.63 |
| Improve | △0.04 | △4.16 | △0.41 | △1.19 | △1.57 | △0.01 | △0.09 |
| CDD [29] | 4.80 | 88.18 | 8.30 | 20.81 | 34.32 | 1.46 | 4.77 |
| CDD+TTD | 4.81 | 88.12 | 9.21 | 23.08 | 37.50 | 1.43 | 4.81 |
| Improve | △0.01 | ▽0.06 | △0.91 | △2.27 | △3.18 | ▽0.03 | ▽0.04 |
| PIAFusion [17] | 4.99 | 79.98 | 8.30 | 21.49 | 31.42 | 0.99 | 6.31 |
| PIAFusion+TTD | 4.85 | 82.09 | 8.43 | 21.90 | 32.45 | 1.38 | 5.76 |
| Improve | ▽0.14 | △2.11 | △0.13 | △0.41 | △1.03 | △0.39 | △0.55 |

# B Implementation Details

## B.1 Algorithm

Here we report the algorithm of the whole dynamic fusion strategy in Algorithm 1. To accomplish our TTD, we initially feed each uni-source image individually into the fusion network to acquire the respective uni-source components. Then we compute the pixel-wise loss between the uni-source components and their corresponding uni-source images. Finally, utilizing Eq. (6) we obtain the dynamic fusion weight and apply dynamic fusion accordingly.

---

**Algorithm 1** algorithm of dynamic fusion strategy

---

**Input:** $x = \left\{ x^{(m)} | m = 1, 2, ..., M \right\}$,
**Output:** $I_F$
1: **for** each $m \in [1, M]$ **do**
2: $\quad \ell^{(m)} = \| D(E^{(m)}(x^{(m)})) - x^{(m)} \|$
3: **end for**
4: **for** each $m \in [1, M]$ **do**
5: $\quad \omega^{(m)} = Softmax(e^{-\ell^{(m)}})$
6: **end for**
7: $I_F = D \left( \sum_{m=1}^{M} \omega^{(m)} \cdot E^{(m)} \left( x^{(m)} \right) \right)$
8: **return** F

---

## B.2 The Pipeline of TTD

The detailed pipeline for inference is shown in Fig. 6 (c). In stage 1 (black dashed line), we feed each uni-source image individually into the frozen encoder and decoder to acquire the respective decomposed uni-source components. Then, we calculate the RDs according to the colored line in Eq. (6) (c). In stage 2 (solid line), we feed multi-source images into the encoder and get their corresponding features. Then, we fuse features by multiplying the RDs to the respective features and adding them up. Finally, the fused feature is fed into the decoder for the final fusion results.

# C Experimental Results

## C.1 Visualization of Relative Dominability

**RD is adaptable to the noise condition.** We simulate a noisy situation in which the visible image quality is affected by contrast perturbation. As shown in Fig. 7, with the corruption severity level increasing, the dominant regions of visible modality are gradually reduced, while the unchanged infrared modality gains an increasing RD. Our RD effectively perceives the dominance changes.

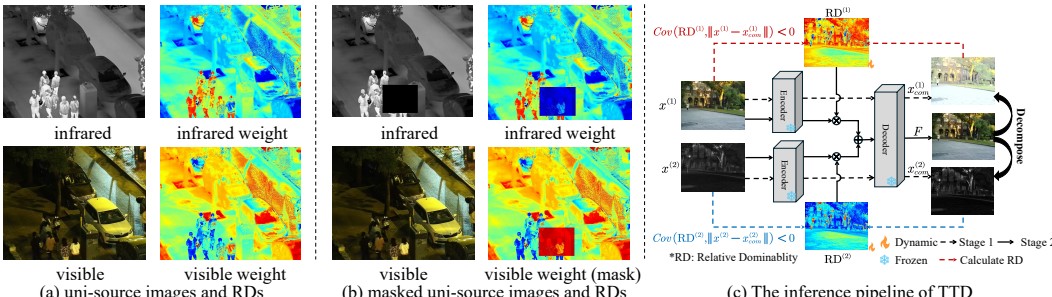

(a) uni-source images and RDs (b) masked uni-source images and RDs (c) The inference pipeline of TTD

Figure 6: (a)(b) Visualization of RDs in mask condition. We randomly masked uni-source data. The RD of the region being masked is apparently smaller than the surrounding area, while that of the same region in the infrared image is relatively greater. (c) The pipeline of TTD

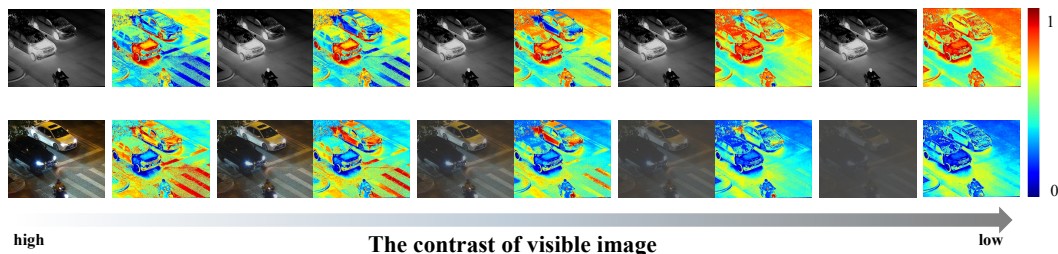

Figure 7: Visualization of RDs with varying contrast visible images. With the corruption severity level (contrast perturbation) increasing, the dominant regions of visible modality are gradually reduced. Our RD effectively perceives the changes on visible modality in the visualizations, while the unchanged infrared modality gains an increasing RD.

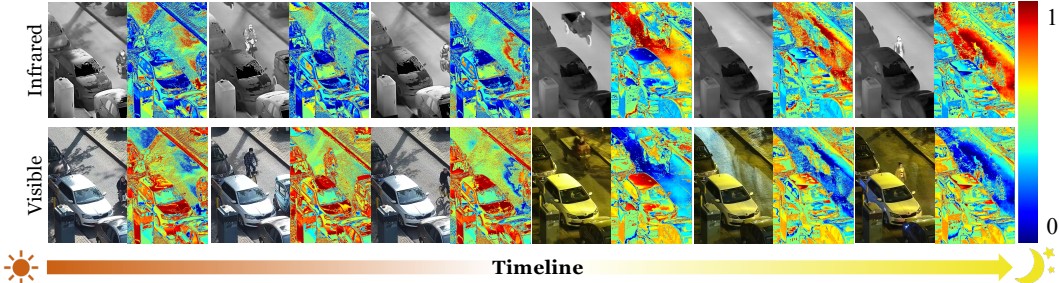

Figure 8: Visualizations of RD maps at different times within the same scenario in the LLVIP dataset. As time progresses from day to night, an intuitive observation is that the dominance of visible images gradually decreases, while the dominance of infrared images increases.

**RD is adaptable to different data qualities.** a) The quality of images also changes with illumination. As shown in Fig. 8, we visualized the RDs of the samples at different times in the same scenario. As it changes from day to night, the dominance of visible images gradually decreases, while the dominance of infrared images increases.

b) Furthermore, to simulate the malfunction of sensors in a real scenario, we masked the infrared image randomly. As shown in Fig. 6 (a)(b), the RD of the region being masked is apparently smaller than the surrounding area, while that of the same region in the infrared image is relatively greater.

## C.2 Ablation Study

As shown in the ablation study results of our TTD on four tasks in Fig. 9, our TTD can highlight the advantageous regions in four tasks, improve contrast, and preserve detail compared with baselines. For example, in the VIF task, our method enhances the details of people and the shadow textures of trees in the fused images compared to the baseline. In the MIF task, our method maintains the bright information from PET while strengthening the texture details from MRI in the overlapping regions. In the MEF and MFF tasks, the fused images produced by our method have stronger texture details and edge information, as well as higher clarity and color fidelity, compared to the baseline fused images.

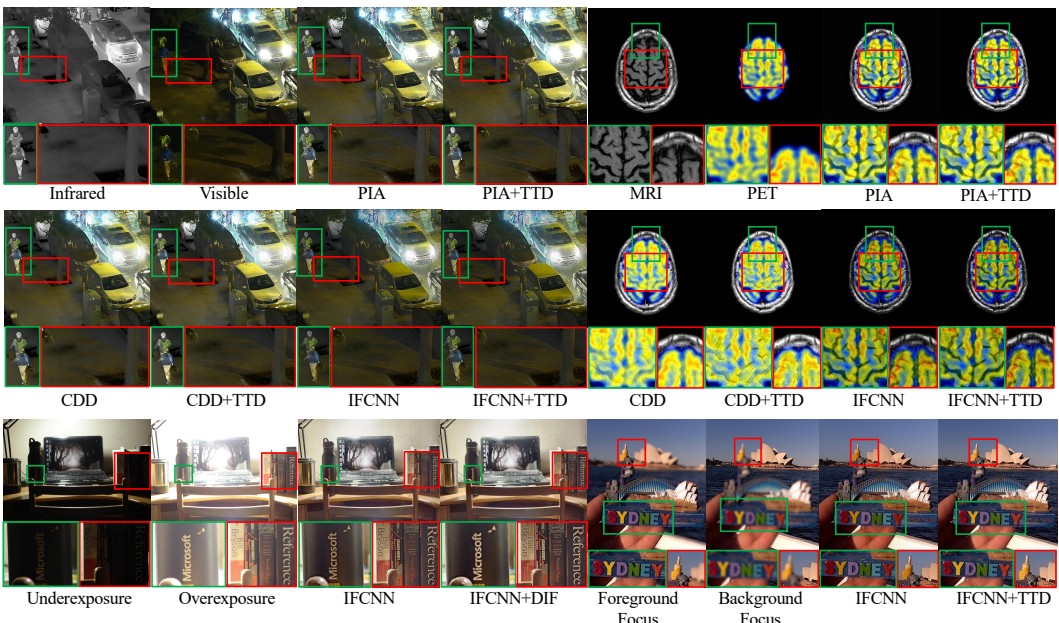

Figure 9: The ablation study of TTD on four tasks.

Table 4: Ablation study on different forms of fusion weights on LLVIP dataset.

| Forms of weight | EN↑ | SD↑ | AG↑ | EI↑ | SF↑ | SSIM↑ | CE↓ |
|---|---|---|---|---|---|---|---|
| $w = 0.5$ | 6.95 | 37.75 | 5.18 | 13.13 | 18.18 | 1.32 | 7.82 |
| $w = Softmax(-\ell)$ | 6.97 | 38.41 | 5.24 | 13.31 | 18.31 | **1.35** | 7.81 |
| $w = Softmax(Sigmoid(-\ell))$ | 6.97 | 38.48 | 5.36 | 13.60 | 18.87 | 1.33 | 7.80 |
| $w = Softmax(e^{-\ell})$ | **6.98** | **38.99** | **5.48** | **13.92** | **19.40** | 1.34 | **7.79** |

Table 5: Ablation study on the normalization of the weights on LLVIP dataset.

| Forms of normalization | EN↑ | SD↑ | AG↑ | EI↑ | SF↑ | SSIM↑ | CE↓ |
|---|---|---|---|---|---|---|---|
| baseline | 6.95 | 37.75 | 5.18 | 13.13 | 18.18 | 1.32 | 7.82 |
| w/o norm | 6.57 | 29.84 | 4.60 | 11.56 | 16.56 | 0.95 | 8.80 |
| Proportional Norm | 6.97 | 38.41 | 5.24 | 13.31 | 18.31 | **1.34** | 7.80 |
| softmax(ours) | **6.98** | **38.99** | **5.48** | **13.92** | **19.40** | 1.34 | **7.79** |

Our TTD is a simple but effective method with a straightforward structure, and we analyze the effectiveness of the TTD from different aspects. We have summarized these ablated analyses here:

(i) ablation study on different baselines: see Sec. 4.2, Tab. 1, and Tab. 2.

(ii) ablation study on the correlation between weight and loss: see Sec. 5.1 and Fig. 5.

(iii) ablation study on the ways to obtain weight: see Sec. 5.3 and Fig. 5.

(iv) ablation study on different forms of fusion weights. We compared different forms of fusion weight: $w = 0.5$ (baseline),$w = Softmax(-\ell)$, $w = Softmax(Sigmoid(-\ell))$, $Softmax(e^{-\ell})$ over IFCNN on the LLVIP dataset, results are given in Tab. 4, it shows that forms of fusion can be flexible to achieve the negative correlation between weight and reconstruction loss.

(v) ablation study on the normalization of the weights: we compared three forms of normalization over IFCNN on the LLVIP, results are given in Tab. 5, indicating that as a premise of the generalization theory (see Thm. 3.1), the normalization of the weights is necessary and the ways to normalize have little impact on our method.

Overall, we performed complete ablation analyses to validate the effectiveness of TTD (i), the necessity of the negative correlation between fusion weight and reconstruction loss (ii), the expandability of ways to obtain fusion weight (iii), the flexibility in the form of weights (iv), the significance of normalization (v).

Table 6: Quantitative comparison on MRI-PET dataset in medical image fusion task.

| Method | EN↑ | SD↑ | AG↑ | EI↑ | SF↑ | SSIM↑ | CE↓ |
|---|---|---|---|---|---|---|---|
| | **MRI-PET Dataset** | | | | | | |
| Densefuse [26] | 3.80 | 49.62 | 4.32 | 11.14 | 15.38 | 1.48 | 3.99 |
| U2Fusion [22] | 4.31 | 51.57 | 5.54 | 14.54 | 19.06 | 0.43 | 19.10 |
| DeFusion [41] | 4.15 | 63.46 | 5.78 | 15.22 | 21.21 | 1.51 | 3.59 |
| SwinFuse [20] | 2.91 | 54.09 | 4.59 | 11.71 | 21.63 | 1.45 | 3.64 |
| MUFusion [43] | 3.68 | 64.75 | 5.71 | 15.59 | 18.51 | 1.43 | 3.61 |
| DDFM [27] | 3.86 | 54.71 | 5.16 | 13.25 | 18.58 | 1.44 | 3.64 |
| TC-MoA [24] | 4.83 | 71.65 | 6.79 | 17.99 | 22.52 | 1.46 | 4.31 |
| IFCNN [18] | 3.96 | 55.29 | 7.21 | 18.21 | 26.69 | 1.52 | 3.66 |
| IFCNN+TTD | **4.00** | **58.48** | **7.80** | **19.77** | **28.97** | **1.52** | **3.63** |
| Improve | △0.04 | △3.19 | △0.59 | △1.56 | △2.28 | △0.00 | △0.03 |
| CDD [29] | 4.28 | 81.33 | 7.68 | 19.74 | 27.84 | 1.50 | 3.57 |
| CDD+TTD | 4.27 | **83.15** | **8.73** | **22.03** | **32.58** | 1.47 | 3.59 |
| Improve | ▽0.01 | △1.82 | △1.05 | △2.29 | △4.74 | ▽0.03 | ▽0.02 |
| PIAFusion [17] | 4.43 | 73.32 | 8.53 | 21.96 | 30.41 | 0.95 | 5.88 |
| PIAFusion+TTD | 4.33 | **75.79** | **8.75** | **22.59** | **31.65** | **1.40** | **5.39** |
| Improve | ▽0.10 | △2.47 | △0.22 | △0.63 | △1.24 | △0.45 | △0.49 |

Table 7: Quantitative comparison on MRI-SPECT dataset in medical image fusion task.

| Method | EN↑ | SD↑ | AG↑ | EI↑ | SF↑ | SSIM↑ | CE↓ |
|---|---|---|---|---|---|---|---|
| | **MRI-SPECT Dataset** | | | | | | |
| Densefuse [26] | 3.61 | 44.36 | 2.78 | 7.13 | 10.74 | 1.59 | 3.79 |
| U2Fusion [22] | 3.89 | 45.30 | 3.96 | 10.18 | 15.67 | 0.43 | 19.17 |
| DeFusion [41] | 3.77 | 49.80 | 3.37 | 8.77 | 13.32 | 1.60 | 3.53 |
| SwinFuse [20] | 2.79 | 44.29 | 2.84 | 7.24 | 14.16 | 1.53 | 3.74 |
| MUFusion [43] | 3.52 | 50.28 | 4.06 | 10.96 | 14.45 | 1.51 | 3.60 |
| DDFM [27] | 3.69 | 47.74 | 3.32 | 8.99 | 13.18 | 1.56 | 3.70 |
| TC-MoA [24] | 4.44 | 58.27 | 4.44 | 11.59 | 16.30 | 1.52 | 4.43 |
| IFCNN [18] | 3.63 | 47.48 | 4.75 | 11.96 | 19.35 | 1.60 | 3.61 |
| IFCNN+TTD | **3.66** | **49.08** | **5.13** | **12.94** | **21.10** | 1.60 | **3.59** |
| Improve | △0.03 | △1.60 | △0.38 | △0.98 | △1.75 | ▽0.00 | △0.02 |
| CDD [29] | 3.90 | 71.58 | 5.21 | 13.28 | 20.70 | 1.58 | 3.65 |
| CDD+TTD | **3.91** | **74.45** | **5.69** | **14.47** | **23.40** | 1.55 | 3.72 |
| Improve | △0.01 | △2.87 | △0.48 | △1.19 | △2.70 | ▽0.03 | ▽0.07 |
| PIAFusion [17] | 4.08 | 61.62 | 5.60 | 14.23 | 21.92 | 0.97 | 6.33 |
| PIAFusion+TTD | 4.01 | **63.85** | **5.68** | **14.43** | **22.56** | **1.46** | **5.68** |
| Improve | ▽0.07 | △2.23 | △0.08 | △0.20 | △0.64 | △0.49 | △0.45 |

## C.3 Comparison On MIF task

Here we provide the quantitative comparison results on MRI-CT, MRI-PET, and MRI-SPECT datasets in Tab. 3, 6, and 7. Our method yields competitive performance on seven evaluation metrics on the three MIF datasets.

## C.4 The Effectiveness of TTD on Baselines with Varying Performaces

In Tab. 1, we have applied TTD to various baselines with different capabilities and all achieved consistent enhancement, TTD can even further improve the performance when combined with current state-of-the-art methods. To further validate that our TTD is effective on models with different performances, we conducted additional experiments to apply TTD on models with varying performance levels by adding random Gaussian noise to the pre-trained model (IFCNN) parameters. The results on the LLVIP dataset are given in Tab. 8, showing that the performance of the baseline decreases with increasing noise added to it. As a comparison, our

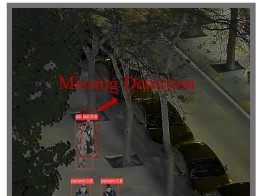 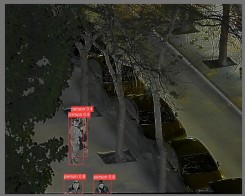 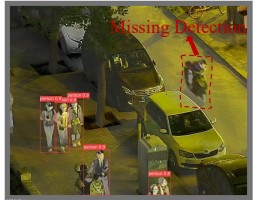 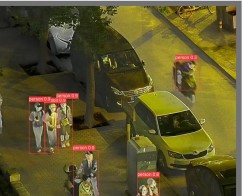

Baseline            Baseline+TTD           Baseline            Baseline+TTD

Figure 10: The comparison of detection results between IFCNN and IFCNN+TTD.

Table 8: The effectiveness of TTD on baselines with varying performances.

| Noise | Method | EN↑ | SD↑ | AG↑ | EI↑ | SF↑ | SCD↑ | CE↓ |
|---|---|---|---|---|---|---|---|---|
| 0.0 | IFCNN | 6.95 | 37.75 | 5.18 | 13.13 | 18.18 | 1.32 | 7.82 |
| | IFCNN+TTD | **6.98** | **38.99** | **5.48** | **13.92** | **19.40** | **1.34** | **7.79** |
| | Improve | △0.03 | △1.24 | △0.30 | △0.79 | △1.22 | △0.02 | △0.03 |
| 0.1 | IFCNN | 6.90 | 36.88 | 5.00 | 12.60 | 18.10 | 1.28 | 7.89 |
| | IFCNN+TTD | **6.92** | **38.01** | **5.26** | **13.27** | **19.33** | **1.30** | **7.87** |
| | Improve | △0.02 | △1.13 | △0.26 | △0.67 | △1.23 | △0.02 | △0.02 |
| 0.2 | IFCNN | 6.86 | 36.61 | 4.90 | 12.34 | 17.44 | 1.28 | 8.15 |
| | IFCNN+TTD | **6.89** | **37.95** | **5.16** | **13.01** | **18.36** | **1.31** | **8.11** |
| | Improve | △0.03 | △1.34 | △0.26 | △0.67 | △0.92 | △0.03 | △0.04 |
| 0.3 | IFCNN | 5.93 | 32.20 | 4.45 | 11.16 | 16.81 | 1.04 | 8.33 |
| | IFCNN+TTD | **5.96** | **33.45** | **4.74** | **11.88** | **18.26** | **1.05** | **8.23** |
| | Improve | △0.03 | △1.25 | △0.29 | △0.72 | △1.45 | △0.01 | △0.10 |
| 0.4 | IFCNN | 6.04 | 32.18 | 4.10 | 10.33 | 15.38 | 1.01 | 8.28 |
| | IFCNN+TTD | **6.33** | **36.81** | **4.69** | **11.89** | **17.48** | **1.11** | **7.99** |
| | Improve | △0.29 | △4.63 | △0.59 | △1.56 | △2.10 | △0.10 | △0.29 |
| 0.5 | IFCNN | 4.49 | 30.38 | 3.72 | 9.33 | 16.03 | 0.75 | 8.99 |
| | IFCNN+TTD | **4.53** | **35.32** | **4.31** | **10.82** | **18.71** | **0.82** | **8.82** |
| | Improve | △0.04 | △4.94 | △0.59 | △1.49 | △2.69 | △0.07 | △0.17 |

Table 9: Per image inference time of TTD on different baselines

| Method | Baselines (s) | Baseline+TTD (s) |
|---|---|---|
| CDDFuse [29] | 1.10 | 1.77 |
| PIAFusion [17] | 2.05 | 4.02 |
| IFCNN [18] | 0.006 | 0.0013 |

TTD effectively improves all these baselines' performance, indicating the effectiveness and generalizability of our TTD on various baselines with different performances.

### C.5 Results on Downstream Task

First, we train the detection model with visible images from the LLVIP dataset. Then we employed our TTD on IFCNN and compared its performance with the baseline on the object detection task. As illustrated in Fig. 10, the detection results of the baseline fused images exhibit missing detection for hard-recognized fast-moving blurred objects. In contrast, after applying our TTD, all objects were accurately detected. Additionally, the fused images obtained using our TTD achieved higher performance in detection tasks compared to the baseline. our DIF shows improvements over the baseline in P, R, and mAP.5:.95 metrics.

## D   Inference Time

The inference time of TTD is dependent on the inference time of the baseline. Since TTD is executed in two stages: in the first stage, we calculate the uni-source reconstruction loss and then compute the fusion weights; in the second stage, we perform the fusion based on the weights. As baselines perform the static fusion, the inference time of TTD is approximately double that of the baseline. We measured the average processing time per image on the test set of the LLVIP dataset. The results of the inference time are given in Tab. 9.

# E    Limitations and Broader Impacts

As a test-time dynamic image fusion method, the performance of our TTD significantly depends on the performance of the baseline models. In the future, we will try to employ the dynamic fusion mechanism in the optimization of baseline models to guide fusion or design a more effective network, further improving fusion performance. Moreover, in the gradient-based TTD, we select the best gradient empirically, a more adaptive selection approach should be explored in the future. As for the potential social impact, our method performs multi-sensor information fusion, which can be applied to drones, cameras, etc., but it is hard to guarantee the effectiveness of baseline models, which may be risky in high-risk scenarios such as medical imaging.

