# OpenReview forum: "Test-Time Dynamic Image Fusion"
_NeurIPS.cc/2024/Conference — NeurIPS 2024 poster_

### Official Review · Reviewer_EtNz · 2024-07-10

**Soundness:** 2
**Presentation:** 2
**Contribution:** 2
**Rating:** 5
**Confidence:** 3

**Summary:**

This paper introduces a method for dynamically adjusting fusion weights for image pixels based on their relative dominability, calculated using pixel-wise reconstruction losses. This approach aims to minimize generalization error by considering the correlation between fusion weights and reconstruction losses.

**Strengths:**

1- The presentation of the method is simple and clear.

2- As the results show, the proposed framework achieves good results on several datasets.

**Weaknesses:**

1. The claim that RD accurately captures the dominant regions of different sources without solid empirical justification for various scenarios might be overreaching. It is unclear how RD performs under various noise conditions or with sources of different qualities.

2. In Eq. (4), the fusion weights $w$ are important as they determine the contribution of each source to the total loss. However, there is no clear definition for normalizing these weights.

3.  In Eq. (8), the paper lacks discussion on the initialization of $w^(m)$  and how it affects convergence and stability during the dynamic adjustment process.

4. Figure 2 is not clear. How are the feature maps of different layers fused, and what is the impact of each of them on the overall performance of the proposed model?

**Questions:**

Please see the weaknesses!

**Limitations:**

Not very adequate. The limitation is general.

---

> ### Author Rebuttal · Authors · 2024-08-07
>
> We’d like to thank the reviewer for the valuable comments, the acknowledgment of our **good results**, and the **simple but clear presentation**. We provide detailed responses to the constructive comments.
> * Weakness 1: More explanations for RD.
>
> Thanks for the valuable comments. We have added experiments under **noise conditions** and with sources of **different qualities**. In addition, our RD's effect was validated on **different methods and tasks**. These experiments and the visualization of RD empirically demonstrate that the proposed dynamic weights (RDs) based on theoretical principles are effective in capturing and emphasizing the dominant regions of different sources, leading to outstanding fusion performance.
>
>   (i) **Comparisons on different tasks over multiple baselines**.
>
> * For different scenarios, we have evaluated our TTD on four image fusion tasks: VIF (see Tab. 1), MIF (see Tab. 3-5), MEF (see Tab. 2), and MFF (see Tab. 2).
> * For different baselines, we have applied our TTD to CDDFuse (CDDFuse+TTD), PIAFusion (PIAFusion+TTD), and IFCNN (IFCNN+TTD), separately. Results are given in Tab. 1-5.
>
>
> (ii) **The visualizations of RDs in various scenarios**.
>
> * **RD is adaptable to the noise condition**. We simulate a noisy situation in which the visible image quality is affected by contrast perturbation. As shown in **Fig. A4**, with the corruption severity level increasing, the dominant regions of visible modality are gradually reduced, while the unchanged infrared modality gains an increasing RD. Our RD effectively perceives the dominance changes.
>
> * **RD is adaptable to different data qualities**.
>     a) To simulate the malfunction of sensors in a real scenario, we masked the infrared image randomly. As shown in **Fig. A2**, the RD of the region being masked is apparently smaller than the surrounding area, while that of the same region in infrared image is relatively greater. b) Furthermore, the quality of images also changes with illumination. As shown in Fig. 6 (see Appendix C.1), we visualized the RDs of the samples at different times in the same scenario. As it changes from day to night, the dominance of visible images gradually decreases, while the dominance of infrared images increases.
> * **RD is adaptable to different tasks**. We have presented visualizations of RDs obtained using CDD+TTD for VIF and MIF tasks, and IFCNN+TTD for MEF and MFF tasks, as CDDFuse is a VIF-specific method while IFCNN is a unified approach. As shown in Fig. 1, our RD reflects the dominance of each source on different tasks adaptively.
>
>
> (iii) **RD's effectiveness over varying performance baseline**. We further conduct additional experiments to combine TTD with baselines of varying performance. As shown in **Tab. A2** of global rebuttal PDF, our TTD significantly improves these baselines with different performance. This validates that our TTD effectively improves baselines in different scenarios and with different performances.
>
> * Weakness 2: More explanations for normalizing the weights.
>
> Thanks for the valuable comment. In Eq. 7, we performed a softmax normalization on $w$ as the sum of $w$ being 1 is a prerequisite for deriving the upper bound of generalization error (GError). As shown in **Fig. A1** of global rebuttal PDF, we normalized the weight maps of different sources at the same positions.
>
> * Weakness 3: More explanations for the initialization.
>
> Thanks for the constructive comment. The initialization of $w$, which is multiplied with unimodal features, **only participates in the uni-source reconstruction process** to adapt the uni-source features to the feature space of the baseline. If the initialization of $w$ is different from that of the baseline in the reconstruction process, **the feature distribution will deviate from the baseline's feature space**, affecting the reconstruction performance. Accordingly, we have added experiments with different initialization weights during uni-source reconstruction to explore the effect of different initializations. According to **Tab. A3** of global rebuttal PDF, we set $w$ the same as the baseline.
>
> Besides, unlike traditional test-time adaptation methods [20][21], our TTD does not require fine-tuning the network. In the fusion process, $w$ can be obtained by a single calculation step (Eq. 6). Theoretically, the fused image can be regarded as a linear combination of $M$ uni-source components. We reveal that the fusion model's upper bound of GError is composed of the distance between the uni-source image and each uni-source component as well as the correlation between fusion weight and uni-source component reconstruction loss according to Eq. 4. As the model is frozen in the test time, the essence of reducing GError lies in the negative correlation between fusion weight and uni-source component reconstruction loss. **TTD performs fusion by the dynamic weight negatively correlated to the reconstruction loss according to Eq. 6 by a single calculation step without any training or fine-tuning**, reducing the generalization error and achieving robust results.
>
> * Weakness 4: More explanations for the framework.
>
> Thanks for your comments. We draw a more detailed pipeline for inference in **Fig. A3**.
>
> **In stage 1** (dashed line), we feed each uni-source image individually into the frozen encoder and decoder to acquire the respective decomposed uni-source components. Then, we construct the RD according to Eq. 6.
>
> **In stage 2** (solid line), we feed multi-source images into the encoder and get their corresponding features. Then, we fuse features by multiplying the RDs to the respective features and adding them up. Finally, the fused feature is fed into the decoder for the final fusion results.
>
> Please kindly note that the fusion of different sources only occurs at the fusion layer of the baseline without interactions between features from different layers. We will release all our code for reliable reproduction.

---

> > ### Comment · Reviewer_EtNz · 2024-08-12
> >
> > The response from the author has addressed my comments. I increased my score.

---

> ### Author Response · Authors · 2024-08-13
>
> Thanks a lot for your reply. We are delighted to have addressed your concerns. We also appreciate your insightful comments, which have greatly improved our work and inspired us to research more.

---

### Official Review · Reviewer_gbUk · 2024-07-12

**Soundness:** 3
**Presentation:** 2
**Contribution:** 3
**Rating:** 7
**Confidence:** 5

**Summary:**

This paper proposes a theoretical justification of image fusion from a generalization perspective and reduces the upper bound of generalization error by decomposing the fused image into multiple components corresponding to its source data. A new test-time dynamic image fusion paradigm TTD is further proposed with the finding that the negative correlation between fusion weight and the uni-source reconstruction loss is the key to reducing the generalization loss. Extensive experiments and discussions confirm the theory and superiority.

**Strengths:**

The idea that applying the test-time adaption method into image fusion task with theoretical guarantee is quite meaningful and experiments are sufficient.

**Weaknesses:**

The details presentation and explanation about test-time adaption are not very clear.

**Questions:**

1. Is the idea that adding up every uni-source data linearly to get fused image reasonable enough? Are there any information only found with several data neglected in the whole process?
2. Mathematical formulas in Appendix A.1 are a little confused. More parentheses ought to be used to present clear explanations about the scopes of every mathematical symbols.
3. Is the TTD applied to every combination from sources? Most test-time adaption methods only need few data to fine-tune the model. But this paper seems to apply TTD to all source data.

**Limitations:**

This paper presents a new perspective to analysis the generalization error of the image fusion task, the details presentation and explanation are not very clear, mathematical formulas are little confused and more explanation about inference process should be provided.

---

> ### Author Rebuttal · Authors · 2024-08-07
>
> We sincerely thank the reviewer for your valuable comments and appreciate your recognition of the **theoretical justification**, **meaningful and sufficient experiments** as well as the **superiority of our work**. We believe the constructive feedback will improve the paper and increase its potential impact on the community.
> * Weakness 1: The details presentation and explanation about test-time adaption are not very clear.
>
> Thanks for your constructive comment. To illustrate the TTD workflow more clearly, we draw a new detailed pipeline during inference in **Fig. A3** (see the global rebuttal PDF).
>
> **In stage 1** (dashed line), we input each uni-source image individually into the frozen encoder and decoder to acquire the respective decomposed uni-source components. Then, we construct the RD which is negatively correlated to the distance between the uni-source component and its original image according to Eq.6.
>
> **In stage 2** (solid line), multiple source images are fed into the encoder at the same time and get their corresponding features. Next, we get the fused feature by multiplying the RDs as weights to their respective features and adding them up. Finally, the fused feature is input into the decoder and the fused image is obtained.
>
> * Question 1: Is the idea that adding up every uni-source data linearly to get fused image reasonable enough? Are there any information only found with several data neglected in the whole process?
>
> Thanks for your valuable comment.
>
>   (i) Most recent approaches [18][19] imply **multi-source image fusion aims at integrating comprehensive information from different source data**. The key challenge lies in capturing the effective component of each uni-source data. To address this problem, we theoretically propose a method that can effectively extract single-source information.
>
> * **Theoretically**, the fused image can be regarded as a linear combination of $M$ uni-source components. We reveal that the model's upper bound of generalization error in the image fusion task is composed of the distance between the uni-source image and each uni-source component as well as the correlation between fusion weight and uni-source component reconstruction loss according to Eq. 4. As the model's encoder and decoder are frozen in the test time, **the essence of reducing generalization error lies in the negative correlation between fusion weights and uni-source component reconstruction loss**. TTD performs fusion by the dynamic weight negatively correlated to the reconstruction loss, achieving a reduction in the generalization error compared with the baseline.
>
> * **Empirically**, the fusion weight, e.g. RD (defined in Eq. 6), since fusion models are trained to extract complementary information from each source, the decomposed components of fusion images represent the effective information from the source data. Thus, the uni-source components can be estimated from source data using the fusion model, with the losses representing the deficiencies of the source in constructing fusion images. Negatively correlated to the reconstruction loss, the **RD effectively demonstrates the dominance of each source and highlights the dominant regions as a fusion weight**.
>
>
> (ii) With the neglect of other source data, the uni-source reconstruction can be regarded as the corresponding decomposed component of the fusion image,  it represents the effective information from the source data, thus the loss between it and source data reflects the Relative Dominability (RD) of the uni-source data in constructing the fusion image, i.e., we **leverage the impact of the missing modality on the loss to perceive the RD** of that source in the fusion image and use the RD as fusion weight. This aligns with the theoretical guarantee of image fusion (see (i)): the key to reducing the generalization error in image fusion tasks lies in the negative correlation between fusion weights and uni-source component reconstruction loss.
> * Question 2: Mathematical formulas in Appendix A.1 are a little confused. More parentheses ought to be used to present clear explanations about the scopes of every mathematical symbols.
>
> Thanks for the valuable suggestions. Here we provide a detailed explanation of the proof in Appendix A.1. At the second equals sign in Eq. 10, the covariance term is derived from the covariance formula: $\mathbb E[XY]=\mathbb E[X]+\mathbb E[Y]+Cov(X, Y)$. At the second inequality sign in Eq. 10, the expression is expanded by grouping $\Vert D(E^{(i)}(x^{(i)})) − x^{(m)}\Vert$ as like terms and expanding the summation notation. Then it is simplified by combining like terms using $w^{(i)}$. At the third inequality sign in Eq. 10, the property of the norm is used, i.e. $\left\|a\right\|-\left\|b\right\|\leq\left\|a+b\right\|$, to achieve further simplification. Due to character limitations, a more detailed derivation and more easily understandable parentheses have been revised and added in Appendix A.1.
> * Question 3: Is the TTD applied to every combination from sources? Most test-time adaption methods only need a few data to fine-tune the model. But this paper seems to apply TTD to all source data.
>
> Thank you for your valuable questions. Most test-time adaptation methods [20][21] require a small amount of data to fine-tune the model. Unlike traditional test-time adaptation methods, our TTD does not require fine-tuning any network parameters. For each sample, the negative correlation between the fusion weight $w$ and the uni-source reconstruction loss can reduce the generalization error upper bound according to the theoretical analysis in the answer to question 2. **TTD performs fusion by the dynamic weight negatively correlated to the reconstruction loss referring to Eq.6 through a single calculation step without any training or fine-tuning**, reducing the generalization error and achieving robust results.

---

> ### Comment · Area_Chair_jXGF · 2024-08-14
> **Reminder for review**
>
> Dear Reviewer gbUk, I have noticed that you have not yet responded to the authors' rebuttal. I kindly urge you to engage in a discussion with the authors at your earliest convenience to help advance the review process.

---

### Official Review · Reviewer_UH2o · 2024-07-13

**Soundness:** 3
**Presentation:** 2
**Contribution:** 3
**Rating:** 7
**Confidence:** 5

**Summary:**

This paper proposes a theoretically guaranteed new paradigm for test-time dynamic image fusion, which exploits the negative correlation between the fusion weights and the single-source reconstruction loss to reduce the upper bound of the generalization error. Extensive experiments demonstrate its effectiveness on a variety of image fusion tasks.

**Strengths:**

The proposed method is simple and effective，the experimental results are detailed and rich, and the effect is competitive compared to SOTAs.

**Weaknesses:**

1. Authors does not mention the change in model inference efficiency.
2. The paper assumes that the decoder of the fusion model is a CNN model, which introduces certain limitations. If the model were a Transformer or Diffusion model, would TDD still be effective?
3. For specific fusion tasks, it is recommended to compare specific methods rather than generalized methods, e.g., multifocus image fusion tasks should compare multifocus image fusion methods.
4. Ablation lacks enough persuasiveness.
5. Table I and II are written inconsistently e.g. TDD and Ours.

**Questions:**

See Weaknesses.

**Limitations:**

See Weaknesses.

---

> ### Author Rebuttal · Authors · 2024-08-07
>
> We would like to thank the reviewer for the thoughtful and thorough comments on our paper as well as for recognizing our **theoretical guarantee**, the **simple but effective framework** of our TTD, the **detailed and rich experiments**, and the **competitive effect compared to SoTAs**. We will also make an effort to increase clarity throughout.
> * Weakness 1: Authors do not mention the change in model inference efficiency.
>
> Thanks for the constructive comment. The inference time of TTD is dependent on the inference time of the baseline. Since TTD is executed in two stages: in the first stage, we calculate the uni-source reconstruction loss and then compute the fusion weights; in the second stage, we perform the fusion based on the weights. As baselines perform the static fusion, **the inference time of TTD is approximately double that of the baseline**. We measured the average processing time per image on the test set of the LLVIP dataset. The results of the inference time over multiple models are given in **Tab. A4** (see the global rebuttal PDF).
> * Weakness 2: The paper assumes that the decoder of the fusion model is a CNN model, which introduces certain limitations. If the model were a Transformer or Diffusion model, would TDD still be effective?
>
> Thanks for the suggestive comment. Although we assume that the decoder of the fusion model is a CNN model, we applied TTD on both **CNN-based** (PIAFusion, IFCNN) and **Transformer-based** (CDDFuse) models, and achieved competitive results, the results are given in Tab. 1 and Tab. 2. In addition, we apply our TTD to a **diffusion-based** image fusion method (Dif-Fusion [1]). The experimental results are given as follows, demonstrating that TTD can be applied to baselines with various network structures and improve their performance on multiple metrics.
>
> | Method | EN | SD | AG | EI | SF | SCD | CE |
> | :------: | :------: |  :------: | :------: | :------: | :------: | :------: | :------: |
> | Dif-Fusion | 7.45 | 50.68 | 3.87 | 10.41 | 13.89 | **1.43** | **7.81** |
> | Dif-Fusion+TTD | **7.45** | **53.52** | **4.75** | **12.80** | **16.62** | 1.31 |  8.14 |
>
> * Weakness 3: For specific fusion tasks, it is recommended to compare specific methods rather than generalized methods, e.g., multifocus image fusion tasks should compare multifocus image fusion methods.
>
> Thanks for the constructive suggestion. We have added comparisons of our TTD with methods specifically applicable to the multi-exposure or multi-focus task, and the results in **Tab. A1** in the global rebuttal PDF shows that our method can outperform these specific methods.
>
> * Weakness 4: Ablation lacks enough persuasiveness.
>
> Thanks for the valuable comment. Our TTD is a simple but effective method with a straightforward structure, and we analyze the effectiveness of the TTD from different aspects in our paper. We have summarized these ablated analyses here:
>
>   (i) Ablation study on **different baselines**: see Sec. 4.2, Tab. 1, and Tab. 2.
>
>   (ii) Ablation study on **the correlation between weight and loss**: see Sec. 5.1 and Fig. 5.
>
>   (iii) Ablation study on **the ways to obtain weight**: see Sec. 5.3 and Fig. 5.
>
>   In addition, we added more ablation experiments here:
>
>   (iv) Ablation study on **different forms of fusion weights**. We compared different forms of fusion weight: $w=0.5$ (baseline),$w = Softmax(-\ell)$, $w = Softmax(Sigmoid(-\ell))$, $Softmax(e^{-\ell})$ over IFCNN on the LLVIP dataset, results are given as follows, it shows that forms of fusion can be flexible to achieve the negative correlation between weight and reconstruction loss.
>
> | Forms of weight | EN    | SD     | AG    | EI     | SF     | SCD   | CE     |
> |-------------|-------|--------|-------|--------|--------|-------|--------|
> | $w=0.5$   | 6.95 | 37.75 | 5.18  | 13.13 | 18.18 | 1.32 | 7.82  |
> | $w=Softmax(-\ell)$    | 6.97 | 38.41  | 5.24 | 13.31 | 18.31 | **1.35** | 7.81  |
> | $w=Softmax(Sigmoid(-\ell))$     | 6.97 | 38.48  | 5.36 | 13.60 | 18.87 | 1.33 | 7.80 |
> |$w=Softmax(e^{-\ell})$       | **6.98** | **38.99**  | **5.48**| **13.92** | **19.40** | 1.34 | **7.79**   |
>
>   (v) Ablation study on **the normalization of the weights**: we compared three forms of normalization over IFCNN on the LLVIP dataset, results are given as follows, indicating that as a premise of the generalization theory (see Theorem 3.1), the normalization of the weights is necessary and the ways to normalize have little impact on our method.
>
> Overall, we performed complete ablation analyses to validate the effectiveness of TTD **(i)**, the necessity of the negative correlation between fusion weight and reconstruction loss **(ii)**, the expandability of ways to obtain fusion weight **(iii)**, the flexibility in the form of weights **(iv)**, the significance of normalization **(v)**.
>
> | Method | EN | SD | AG | EI | SF | SCD | CE |
> | ------------------- | ------- | -------- | ------- | -------- | -------- | ------- | -------- |
> | baseline | 6.95 | 37.75 | 5.18 | 13.13 | 18.18 | 1.32 | 7.82 |
> | w/o norm | 6.57 | 29.84 | 4.60 | 11.56 | 16.56  | 0.95 | 8.80 |
> | Proportional Norm | 6.97 | 38.41 | 5.24 | 13.31 | 18.31 | **1.34** | 7.80 |
> | softmax(ours) | **6.98** | **38.99** | **5.48** | **13.92** | **19.40** | 1.34 | **7.79** |
>
> * Weakness 5: Table I and II are written inconsistently e.g. TDD and Ours.
>
> Thanks for the detailed comments. We have modified the claims to avoid inaccurate descriptions, and we have thoroughly reviewed the entire manuscript and corrected these errors.

---

> > ### Comment · Reviewer_UH2o · 2024-08-11
> >
> > Thanks for the rebuttal. The authors have addressed my concern. I intend to increase my score.

---

> ### Author Response · Authors · 2024-08-11
>
> Thanks a lot for your positive feedback. Your insightful comments have greatly improved our work. We sincerely appreciate your support.

---

### Official Review · Reviewer_gBb1 · 2024-07-14

**Soundness:** 3
**Presentation:** 4
**Contribution:** 2
**Rating:** 5
**Confidence:** 3

**Summary:**

- This paper tries to solve the image fusion task, where multi-source images are provided and one needs to extract and integrate effective information from them.
- The paper demonstrates its effectiveness on four different tasks: VIF, MIF, MEF, and MFF.
- The paper proposes a test-time dynamic image fusion method with theoretical justification.
- This paper theoretically proves the superiority of dynamic image fusion over static image fusion, and provides a generalization error upper bound.
- By using the relative domainability of each source as the dynamic fusion weight, it is able to theoretically improve the generalization of the image fusion model and dynamically emphasize the dominant regions of each source.
- This method theoretically and empirically demonstrates superiority over static fusion methods through extensive experiments on various datasets, including visible-infrared, medical, multi-exposure, and multi-focus image fusion tasks.

**Strengths:**

- The paper is well written.
- The method is evaluated on four different tasks.
- The approach is fairly simple.
- The approach is justified theoretically
- When the baseline model is robust, it can improve the fusion performance.

**Weaknesses:**

- The adaptation method heavily relies on the performance of the baseline model, which can be ineffective when the model performance is poor.
- The improvement over the non-adaptive baseline is minor.

**Questions:**

-

**Limitations:**

-

---

> ### Author Rebuttal · Authors · 2024-08-07
>
> We thank the reviewer for recognizing our **effectiveness on multiple tasks**, **theoretical guarantee**, and **well presentation**. We appreciate your support and constructive suggestions and address your concerns as follows.
> * Weakness 1: The adaptation method heavily relies on the performance of the baseline model, which can be ineffective when the model performance is poor.
>
> Thanks for your comment. Please kindly note that our performance is related to the baseline model as we claimed in the Limitations (Line 479-Line 480), however,  it does not imply that our approach would be ineffective on weak-performance baselines. Based on the generalization theory, our TTD explicitly reduces the upper bound of the generalization error of models regardless of the baselines' performance. We have also added experiments to further clarify our effectiveness on the baseline with different performances.
>   We have elaborated on this in two facts:
>
>   (i) **Theoretically**, referring to the deduce in Sec. 3, the fused image can be regarded as a linear combination of multiple uni-source components, i.e. the uni-source reconstruction. We reveal that the model's upper bound of generalization error in the image fusion task is composed of the distance between the uni-source image and each uni-source component as well as the correlation between fusion weight and uni-source component reconstruction loss according to Eq. 4. As the model's encoder and decoder are frozen in the test time, the essence of reducing generalization error lies in the negative correlation between fusion weights and uni-source component reconstruction loss. In static image fusion (baseline), the correlation between weight and reconstruction loss is 0. In contrast, **the dynamic weight in our TTD is negatively correlated to the reconstruction loss**, achieving a reduction in the generalization error compared with the baseline and improving the baseline's performance effectively. Thus, **the key to TTD functioning is independent of the performance of the baseline theoretically**.
>
>   (ii) **Experimentally** , in our paper, we have applied TTD to various baselines with different capabilities and all achieved consistent enhancement, TTD can even further improve the performance when combined with current state-of-the-art methods. To further validate that our TTD is effective on models with different performances, we conducted additional experiments to apply TTD on models with varying performance levels by adding random Gaussian noise to the pre-trained model (IFCNN) parameters. The results on the LLVIP dataset are given in **Tab. A2** (see the global rebuttal PDF), showing that the performance of the baseline decreases with increasing noise added to it. As a comparison, **our TTD effectively improves all these baselines' performance, indicating the effectiveness and generalizability of our TTD on various baselines with different performances**.
> * Weakness 2 : The improvement over the non-adaptive baseline is minor.
>
> Thanks for your comment. Please kindly note that our method is independent of the baseline's adaptability, which of TTD is reflected in its ability to dynamically adjust the fusion weights to each sample.
>
>   (i) **The adaptability of TTD**. As analyzed in Question 1, TTD can theoretically reduce the image fusion model's generalization error by constructing the negative correlation between fusion weight and uni-source component reconstruction loss. For each sample, we derive a pixel-level Relative Dominablity (RD) as the dynamic fusion weight, and RD is negatively correlated with uni-source component reconstruction loss according to Eq. 6. Experimental results in Tab. 1-5, the visualization in Fig. 1 and Fig. 6 show that RD-based dynamic fusion weight effectively captures the dominance of each source in image fusion and enhances its advantages in the fused images. Overall, **this adaptability of TTD refers to its adaptive fusion weights instead of the baseline**.
>
>   (ii) **The improvements over various baselines are not minor**. We applied our TTD on three baselines with different performances, and we also conducted extensive experiments on multi-modal, multi-exposure, and multi-focus datasets. The superior performance across diverse metrics demonstrates the effectiveness and applicability of our approach. We further perform additional experiments to combine TTD with baseline models of varying performance. As shown in **Tab. A2** (see the global rebuttal PDF), our TTD significantly improves these baselines with different performances. **This validates that our TTD effectively improves the various baselines in different scenarios as well as with different performances**.

---

> ### Comment · Area_Chair_jXGF · 2024-08-14
> **Reminder for review**
>
> Dear Reviewer gBb1, I have noticed that you have not yet responded to the authors' rebuttal. I kindly urge you to engage in a discussion with the authors at your earliest convenience to help advance the review process.

---

### Author Rebuttal · Authors · 2024-08-07

Dear PCs, SACs, ACs, and Reviewers,

We would like to thank you for your valuable feedback and insightful reviews, which have greatly contributed to improving the paper. This is a **clear and well-written** (Reviewer gBb1, Reviewer EtNz) manuscript with a **theoretical guarantee** (Reviewer gBb1, Reviewer UH2o, Reviewer gbUk), we proposed a **effective and superior framework** (Reviewer UH2o, Reviewer gbUk), **the meaningful, detailed and sufficient experiments** on multiple tasks validate the theory and **TTD’s effectiveness and superiority** (Reviewer UH2o, Reviewer gbUk, Reviewer gBb1, Reviewer EtNz).

In our rebuttal, we addressed the following raised concerns/misunderstandings.

  * We have provided a detailed explanation and experimental validation for TTD's performance on baselines with varying performance.

  * We have provided the inference time of TTD and baselines.

  * We have validated TTD's adaptability on Dif-Fusion.

  * We have compared our TTD with MEF-specific methods and MFF-specific methods.

  * We have added more ablation experiments to verify the flexibility in the form of weights and the significance of normalization.

  * We have drawn a new detailed pipeline during inference in Fig. A3.

  * We have visualized the RDs under different noise conditions and with sources of different qualities.

  * We have conducted an ablation study on the initialization of $w$.

We hope that our responses will satisfactorily address your questions and concerns. We sincerely appreciate the time and effort you have dedicated to reviewing our submission, along with your invaluable suggestions. We believe that these clarifications and additional details strengthen our paper and address the reviewers' concerns. We understand the constraints of time and workload that reviewers and AC face, and we appreciate the effort already put into evaluating our work. If there are any additional insights, questions, or clarifications on our responses/submission that you would like to discuss with us, we would be very grateful to hear them, your feedback is invaluable for the improvement of our research.


Best regards,

Authors of Submission 29


## Reference
[1] Yue J, Fang L, Xia S, et al. Dif-fusion: Towards high color fidelity in infrared and visible image fusion with diffusion models[J]. IEEE Transactions on Image Processing, 2023.

[2] Ram Prabhakar K, Sai Srikar V, Venkatesh Babu R. Deepfuse: A deep unsupervised approach for exposure fusion with extreme exposure image pairs[C]//Proceedings of the IEEE international conference on computer vision. 2017: 4714-4722.

[3] Wang Q, Chen W, Wu X, et al. Detail-enhanced multi-scale exposure fusion in YUV color space[J]. IEEE Transactions on Circuits and Systems for Video Technology, 2019, 30(8): 2418-2429.

[4] Liu Y, Wang Z. Dense SIFT for ghost-free multi-exposure fusion[J]. Journal of Visual Communication and Image Representation, 2015, 31: 208-224.

[5] Lee S, Park J S, Cho N I. A multi-exposure image fusion based on the adaptive weights reflecting the relative pixel intensity and global gradient[C]//2018 25th IEEE international conference on image processing (ICIP). IEEE, 2018: 1737-1741.

[6] Li H, Zhang L. Multi-exposure fusion with CNN features[C]//2018 25th IEEE International Conference on Image Processing (ICIP). IEEE, 2018: 1723-1727.

[7] Ma K, Duanmu Z, Yeganeh H, et al. Multi-exposure image fusion by optimizing a structural similarity index[J]. IEEE Transactions on Computational Imaging, 2017, 4(1): 60-72.

[8] Lei J, Li J, Liu J, et al. GALFusion: Multi-exposure image fusion via a global–local aggregation learning network[J]. IEEE Transactions on Instrumentation and Measurement, 2023, 72: 1-15.

[9] Liu J, Wu G, Luan J, et al. HoLoCo: Holistic and local contrastive learning network for multi-exposure image fusion[J]. Information Fusion, 2023, 95: 237-249.

[10] Li J, Guo X, Lu G, et al. DRPL: Deep regression pair learning for multi-focus image fusion[J]. IEEE Transactions on Image Processing, 2020, 29: 4816-4831.

[11] Amin-Naji M, Aghagolzadeh A, Ezoji M. Ensemble of CNN for multi-focus image fusion[J]. Information Fusion, 2019, 51: 201-214.

[12] Xu H, Fan F, Zhang H, et al. A deep model for multi-focus image fusion based on gradients and connected regions[J].

[13] Qiu X, Li M, Zhang L, et al. Guided filter-based multi-focus image fusion through focus region detection[J]. Signal Processing: Image Communication, 2019, 72: 35-46.

[14] LAI R U I, LI Y, GUAN J, et al. Multi-Scale Visual Attention Deep Convolutional Neural Network for Multi-Focus Image Fusion[J].

[15] Song X, Wu X J. Multi-focus image fusion with PCA filters of PCANet[C]. Multimodal Pattern Recognition of Social Signals in Human-Computer-Interaction: 5th IAPR TC 9 Workshop, MPRSS 2018, Beijing, China, August 20, 2018, Revised Selected Papers 5. Springer International Publishing, 2019: 1-17.

[16] Ma B, Zhu Y, Yin X, et al. Sesf-fuse: An unsupervised deep model for multi-focus image fusion[J]. Neural Computing and Applications, 2021, 33: 5793-5804.

[17] Ma J, Zhou Z, Wang B, et al. Multi-focus image fusion using boosted random walks-based algorithm with two-scale focus maps[J]. Neurocomputing, 2019, 335: 9-20.

[18] Xu H, Ma J, Jiang J, et al. U2Fusion: A unified unsupervised image fusion network[J]. IEEE Transactions on Pattern Analysis and Machine Intelligence, 2020, 44(1): 502-518.

[19] Tang L, Yuan J, Zhang H, et al. PIAFusion: A progressive infrared and visible image fusion network based on illumination aware[J]. Information Fusion, 2022, 83: 79-92.

[20] Liu Y, Kothari P, Van Delft B, et al. Ttt++: When does self-supervised test-time training fail or thrive?[J]. Advances in Neural Information Processing Systems, 2021, 34: 21808-21820.

[21] Sun Y, Wang X, Liu Z, et al. Test-time training with self-supervision for generalization under distribution shifts[C]. International conference on machine learning. PMLR, 2020: 9229-9248.

---

### Decision · Program_Chairs · 2024-09-25

**Decision:**

Accept (poster)

**Comment:**

The final rating for this paper is two borderline accrpt and two accept. Overall, the reviewers' opinions on the paper were consistent and generally positive. Moreover, most reviewers indicated that the authors had addressed their previous concerns in the rebuttal. Additionally, two reviewers raised their scores after the rebuttal phase. In summary, the overall evaluation of the work is positive, and the reviewers found the authors' responses to be proactive and effective. Therefore, I am inclined to accept this work.